Manuscript prepared for J. Name
with version 2015/04/24 7.83 Copernicus papers of the LATEX class copernicus.cls.
Date: 24 May 2017

# Source apportionment of NMVOCs in the Kathmandu Valley during the SusKat-ABC international field campaign using positive matrix factorization

Chinmoy Sarkar[1], Vinayak Sinha[1], Baerbel Sinha[1], Arnico K. Panday[2], Maheswar Rupakheti[3,4], and Mark G. Lawrence[3]

[1]Department of Earth and Environmental Sciences, Indian Institute of Science Education and Research (IISER) Mohali, Sector 81, S. A. S. Nagar, Manauli PO, Punjab, 140306, India
[2]International Centre for Integrated Mountain Development (ICIMOD), Khumaltar, Lalitpur, Nepal
[3]Institute for Advanced Sustainability Studies (IASS), Berliner Str. 130, 14467 Potsdam, Germany
[4]Himalayan Sustainability Institute (HIMSI), Kathmandu, Nepal

*Correspondence to:* V. Sinha (vsinha@iisermohali.ac.in)

**Abstract.** A positive matrix factorization model (US EPA PMF version 5.0) was applied for the source apportionment of the dataset of 37 NMVOCs measured over a period of 19 December 2012 – 30 January 2013 during the SusKat-ABC international air pollution measurement campaign using a Proton Transfer Reaction Time of Flight Mass Spectrometer in the Kathmandu Valley. In all,

eight source categories were identified with the PMF model using the new "constrained model operation" mode. Unresolved industrial emissions and traffic source factors were the major contributors to the total measured NMVOC mass loading (17.9 % and 16.8 %, respectively) followed by mixed industrial emissions (14.0 %), while the remainder of the source was split approximately evenly between residential biofuel use and waste disposal (10.9 %), solvent evaporation (10.8 %), biomass

co-fired brick kilns (10.4 %), biogenic emissions (10.0 %) and mixed daytime factor (9.2 %). Conditional probability function (CPF) analyses were performed to identify the physical locations associated with different sources. Source contributions to individual NMVOCs showed biomass co-fired brick kilns significantly contribute to the elevated concentrations of several health relevant NMVOCs such as benzene. Despite the highly polluted conditions, biogenic emissions had largest contribution

(24.2 %) to the total daytime ozone production potential, even in winter, followed by solvent evaporation (20.2 %), traffic (15.0 %) and unresolved industrial emissions (14.3 %). Secondary organic aerosol (SOA) production had approximately equal contributions from biomass co-fired brick kilns (28.9 %) and traffic (28.2 %). Comparison of PMF results based on the in-situ data versus REAS v2.1 and EDGAR v4.2 emission inventories showed that both the inventories underestimate the

contribution of traffic and do not take the contribution of brick kilns into account. In addition, the REAS inventory overestimates the contribution of residential biofuel use and underestimates the contribution of solvent use and industrial sources in the Kathmandu Valley. The quantitative source

apportionment of major NMVOC sources in the Kathmandu Valley based on this study will aid in improving hitherto largely un-validated bottom up NMVOC emission inventories, enabling more 25 focused mitigation measures and improved parameterizations in chemical-transport models.

## 1 Introduction

Non-methane volatile organic compounds (NMVOCs) are important atmospheric constituents and are emitted from both natural and anthropogenic sources (Hewitt, 1999). They are important as precursors of surface ozone and secondary organic aerosols (SOA) and affect atmospheric oxidation 30 capacity, climate and human health (IPCC, 2013). Thus, identification of NMVOC sources is necessary for devising appropriate mitigation strategies to improve air quality and reduce undesired impacts of secondary pollutants such as tropospheric ozone and secondary organic aerosol.

Source apportionment of NMVOCs can be achieved by applying source–receptor models to measured ambient datasets. Ambient NMVOC mixing ratios depend on the emission profiles of the 35 sources contributing to the ambient mixture, their relative source strengths, transport, mixing and removal processes in the atmosphere. Source receptor models perform statistical analyses on the dataset to identify and quantify the contribution of different sources to the measured NMVOC concentrations (Watson et al., 2001). Positive matrix factorization (PMF) is currently among the most widely applied receptor models for the source apportionment of NMVOCs, in particular for datasets 40 with high temporal resolution (Anderson et al., 2002; Miller et al., 2002; Kim et al., 2005; Buzcu and Fraser., 2006; Brown et al., 2007; Vlasenko et al., 2009; Slowik et al., 2010; Yuan et al., 2012; Crippa et al., 2013; Kaltsonoudis et al., 2016). In comparison to other receptor models based on principal component analysis/absolute principal component scores (PCA/APCS) (Guo et al., 2004, 2006), chemical mass balance (CMB) (Na and Pyo Kim., 2007; Morino et al., 2011) and UNMIX (Jorquera 45 and Rappenglück., 2004; Olson et al., 2007), PMF provides more robust results as it does not permit negative source contributions. Moreover, a priori knowledge about the number and signature of NMVOC source profiles are not required, which is particularly useful and apt for NMVOC source apportionment studies in a new or understudied atmospheric chemical environment. The recently developed PMF version 5.0 also allows further refining the solution and reducing rotational ambiguity 50 of the solutions using pre-existing knowledge of emission ratios from known point sources. Source apportionment of non-methane hydrocarbons (NMHCs) and oxygenated VOCs (OVOCs) using PMF source–receptor models has been carried out in several previous studies (Shim et al., 2007; Leuchner and Rappenglück , 2010; Gaimoz et al., 2011; Bon et al., 2011; Chen et al., 2014).

NMVOC emission inventories are frequently associated with large uncertainties (Zhang et al., 55 2009). This is particularly true for metropolitan cities in the developing world. Emission inventories can be evaluated using the results obtained from source receptor models such as the PMF model. This evaluation is important to improve the accuracy of the existing emission inventories and therefore to

develop effective air pollution control strategies. In this study, we report the application of the PMF model for source apportionment of NMVOCs using the NMVOC data measured in the Kathmandu Valley, Nepal, which has been reported and analyzed in detail in Sarkar et al. (2016).

Kathmandu is considered to be amongst the most polluted cities in Asia (Panday et al., 2009). According to the existing Nepalese emission inventory (International Centre for Integrated Mountain Development's (ICIMOD) database) and the REAS v2.1 (Kurokawa et al., 2013) emission inventories residential biofuel use is considered to be the most important anthropogenic source of NMVOCs in the Kathmandu Valley. It is considered to contribute $\sim 67\,\%$ (REAS) to $\sim 83\,\%$ (Nepalese inventory), towards the total NMVOC mass loadings. In contrast,EDGAR v4. (Olivier et al., 1994) attributes 66 % of the emissions in the Kathmandu Valley to solvent use and a recent emission inventory study conducted by the International Centre for Integrated Mountain Development (ICIMOD) which relied on measurement of particulate matter (Figure S7) suggested that traffic is the dominant source (69 %) of air pollution in a part of the Kathmandu Valley within the Ring Road (i.e. the Kathmandu Metropolitan City (KMC) and Lalitpur Sub-metropolitan City) and some nearby sub-urban rural areas outside the Ring Road (Pradhan et al., 2012).

The objective of the current study is to identify and quantify the contributions of different emission sources to the ambient wintertime NMVOC concentrations in the Kathmandu Valley using a positive matrix factorization (US EPA PMF 5.0; Brown et al. (2015)) receptor model. NMVOC measurements were carried out at Bode, a suburban site in the Kathmandu Valley over a period of 19 December 2012 – 30 January 2013 during the SusKat-ABC field campaign. The NMVOC measurements, new findings and qualitative analyses of sources have been presented and discussed in Sarkar et al. (2016). The NMVOC measurements suggested significant contribution of varied emission sources such as traffic (associated with high toluene, xylenes and trimethylbenzenes), biomass co-fired brick kilns (associated with high acetonitrile and benzene), industries and wintertime biogenic sources (as characterized by high daytime isoprene). Based on the NMVOCs emission profiles, two distinct periods were identified in the dataset: the first period (19 December 2012 – 3 January 2013) was associated with high daytime isoprene concentrations whereas the second period (4 – 18 January 2013) was associated with sudden increase in acetonitrile and benzene concentrations which was attributed to the start in operations of biomass co-fired brick kilns in the Kathmandu Valley (Sarkar et al., 2016). For quantitative source apportionment, hourly mean measured concentrations of all 37 NMVOCs measured during the instrumental deployment (19 December 2012 – 30 January 2013), were used for the PMF analysis. Sensitivity tests were conducted for the PMF 5.0 model version to evaluate how the new rotational tool called "constrained model operation feature" improves the representation of source profiles in the PMF model output. To identify the physical locations for the identified sources, an important prerequisite for targeted mitigation, conditional probability function (CPF) analyses were also performed. The results obtained from the PMF analyses were compared with three emission inventories – the existing Nepalese inventory, REAS v2.1 (Regional Emission

inventory in ASia) and the EDGAR v4.2 (Emissions Database for Global Atmospheric Research) emission inventory. Additionally, the contributions of each source category to individual NMVOC mass concentrations, ozone formation potential and formation of secondary organic aerosol (SOA) were also analyzed.

## 2  Materials and Methods

### 2.1  Site Description

NMVOC measurements during this study were performed in the winter season from 19 December 2012 until 30 January 2013 at Bode (27.689° N, 85.395° E, 1345 m a.m.s.l.) in Bhaktapur district, which is a suburban site located in the westerly outflow of the Kathmandu Metropolitan City. The land use in the vicinity of the measurement site consisted of the following cities - Kathmandu
Metropolitan City ($\sim 10$ km to the west), Lalitpur Sub-Metropolitan City ($\sim 12$ km south-west of the site) and Bhaktapur Municipality ($\sim 5$ km south-east of the site). The site is located in the Madhyapur-Thimi Municipality. In addition, the region north of the site had a small forested area (Nilbarahi Jungle; $\sim 0.5$ km$^2$ area) and a reserve forest (Gokarna Reserve Forest; $\sim 1.8$ km$^2$ area) at approximately 1.5 km and 7 km from the measurement site, respectively. Several brick kilns were
located in the south-east of the site within a distance of 1 km. Major industries were located mainly in the Kathmandu and Patan cities whereas Bhaktapur industrial estate was located at around 2 km from the measurement site (in the south-eastern direction). A substantial number of small industries were also located in the south-eastern direction. The Tribhuvan International Airport is located about 4 km to the west of the Bode site. A detailed description of the measurement site and prevalent me-
teorology is already provided in the companion paper to this special issue Sarkar et al. (2016). A zoomed view of the land use in the vicinity of the measurement site is provided in Figure 1.

### 2.2  PTR-TOF-MS measurements

NMVOC measurements were performed using a high-sensitivity PTR-TOF-MS (model 8000; Ionicon Analytic GmbH, Innsbruck, Austria) over a mass range of 21-210 amu. The PTR-TOF-MS
instrument works on the basic principle of soft chemical ionization (CI) where reagent hydronium ions (H3O$^+$) react with analyte NMVOC molecules having proton affinity (P.A) greater than that of water vapour (165 Kcal/mol) to form protonated molecular ions (with m/z ratio = molecular ion + 1), enabling the identification of NMVOCs (Lindiger et al., 1998). As all the relevant analytical details pertaining to the PTR-TOF-MS instrument, ambient air sampling and the quality assurance of
the NMVOC dataset has already been provided in detail in Sarkar et al. (2016), only a brief description of the ambient air sampling and the analytical operating conditions is provided here. Ambient air sampling was performed continuously through a Teflon inlet line protected from floating dust and debris using an in-line Teflon membrane particle filter. The PTR-TOF-MS was operated at a drift

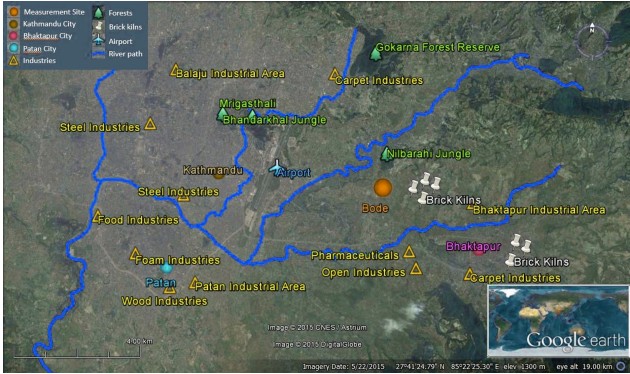

**Figure 1.** Location of the measurement site (Bode, orange circle) along with surrounding cities (Kathmandu, brown circle; Patan, turqoise circle; Bhaktapur, pink circle), brick kilns (white markers), major industries (yellow triangles), forest areas (green tree symbols), airport (blue marker) and major river paths (sky blue) in the Google Earth image of the Kathmandu Valley (obtained on 22 May 2015 at 14:55 LT).

tube pressure of $2.2\,\mathrm{mbar}$, a drift tube temperature of $60°\mathrm{C}$ and a drift tube voltage of $600\,\mathrm{V}$ which
resulted in an operating E/N ratio of $\sim135$ Td (E = electrical field strength in $\mathrm{V\,cm^{-1}}$; N = buffer gas number density in $\mathrm{molecule\,cm^{-3}}$; 1 Td = $10^{-17}\,\mathrm{V\,cm^{-2}}$). Identification of several previously unmeasured and rarely measured NMVOCs were achieved due to the high mass resolution (m/$\Delta$m > 4000) and low detection limit (few tens of ppt) of the instrument. For the quality assurance of the measured NMVOC dataset, the instrument was calibrated twice during the measurement period
and regular instrumental background checks were performed using zero air at frequent intervals. Detailed description of the sensitivity characterization of the instrument and the quality assurance of the primary dataset is available in Sarkar et al. (2016).

During the measurement period, a total of 37 NMVOC signals (m/z) were observed in the PTR-TOF-MS mass spectra that had an average concentration of > 200 ppt. The cut-off of an average
concentration of > 200 ppt was employed keeping in mind the highest instrumental background signals observed during the campaign, so as to have complete confidence that the ions signals were attributable to ambient compounds. For mass identifications at a particular m/z ratio, further quality control was applied. Firstly, only those ion peaks were considered for the mass assignments for which there were no contribution from the major shoulder ion peaks within a mass width bin of
0.005 amu. Next, ion peaks devoid of any variability (that is the time series profile was flat) were not considered for mass assignments at all. Further details including some known interferences that were identified and taken into account are available in Sarkar et al. (2016). Table S1 in the supplementary information lists the identified 37 NMVOCs the corresponding m/z attributions (with references to few previous works which reported the same compound assignment, wherever applicable), and the
elemental molecular formula.

### 2.3 Collection of grab samples

Grab samples from garbage fires (termed garbage burning) were collected near the measurement site ($\sim 200\,\text{m}$ in the northern direction, upwind of Bode; 27.690° N, 85.395° E) on 7 December 2014 between 15:00 - 15:03 LT. A "brick kiln" grab sample was collected on 6 December 2014 from a fixed chimney bull's trench brick kiln (FCBTK) co-fired using coal, wood dust and sugarcane extracts. Figure S1 of the supplementary information shows pictures of the grab sample collection and the instrumental setup for the analysis. The whole air samples were collected in 2 litre glass flasks that had been validated for the stability of NMVOCs (Chandra et al., 2017) and were analyzed within 38 hours of the collection (on 9 December 2014 between 03:42 - 04:05 LT). The whole air samples were diluted (dilution factor of 9.93) using zero air for the quantification of NMVOCs present in the grab samples using a PTR-QMS instrument (Sinha et al., 2014). The average background signals (zero air) were subtracted from each $m/z$ channel and stable data of at least 10 cycles ($\sim 10\,\text{minutes}$) were considered for the calculation of mixing ratios as per the protocol described by Sinha et al. (2014). The zero air background for the m/z reported was 0.04±0.05 ppb, 0.04±0.04 ppb, 0.04±0.06 ppb, 0.07±0.08 ppb, 0.10±0.11 ppb, 0.02±0.06 ppb and 0.02±0.05 ppb for acetonitrile, benzene, toluene, sum of C8 aromatics, sum of C9 aromatics, styrene and naphthalene, respectively. The concentration range in the grab samples was 4±0.3 to 323±8 ppb for acetonitrile, 27±4 to 339±19 ppb for benzene, 32±5 to 150±14 ppb for toluene, 40±6 to 113±8 ppb for C8 aromatics, 33±6 to 62±12 ppb for C9 aromatics, 11±1.3 to 95±17 ppb for styrene and 11±1.5 to 64±9 ppb for naphthalene.

### 2.4 Positive Matrix Factorization (PMF)

The United States Environmental Protection Agency's (US EPA) Positive Matrix Factorization (PMF) receptor model version 5.0 (Norris et al., 2014) was used for source apportionment of NMVOCs in the Kathmandu Valley. The model is based on the multi-linear engine (ME-2) approach and has been described in detail by Paatero (1997, 1999). From a data matrix of a number of NMVOCs in a given number of samples, the PMF model helps to determine the total number of possible NMVOC source factors, the chemical fingerprint (source profile) for each factor, the contribution of each factor to each sample, and the residuals of the dataset using the following equation (Paatero and Tapper , 1994),

$$X_{ij} = \sum_{k=1}^{p} g_{ik} f_{kj} + e_{ij} \tag{1}$$

Where, $X_{ij}$ is the NMVOC data matrix with $i$ number of samples and $j$ number of measured NMVOCs which are resolved by the PMF to provide $p$ number of possible source factors with the source profile $f$ of each source and mass $g$ contributed by each factor to each individual sample, leaving the residuals $e$ for each sample. To obtain the solution of equation (1), sum of the squared

residuals ($e^2$) and variation of data points ($\sigma^2$) are inversely weighted in PMF as expressed by the following equation (Paatero and Tapper , 1994),

$$Q = \sum_{i=1}^{n}\sum_{j=1}^{m}(\frac{e_{ij}}{\sigma_{ij}})^2 = \sum_{i=1}^{n}\sum_{j=1}^{m}(\frac{X_{ij} - \sum_{k=1}^{p}g_{ik}f_{kj}}{\sigma_{ij}})^2 \tag{2}$$

Where, $Q$ is the object function and a critical parameter for PMF, $n$ is the number of samples, and $m$ is the number of considered species. The original data should always be reproduced by the PMF model within the uncertainty considering the non-negativity constraint for both the predicted source profile and the predicted source contributions. The explained variability (EV) as given below demonstrates the relative contribution of each factor to the individual compound and can be expressed as (Gaimoz et al., 2011),

$$EV_{kj} = \frac{\sum_{i=1}^{n}|g_{ik}f_{kj}|/\sigma_{ij}}{\sum_{i=1}^{n}(\sum_{k=1}^{p}|g_{ik}f_{kj}| + |e_{ij}|)/\sigma_{ij}} \tag{3}$$

The explained variability is most useful to policy makers. If the observed mass loading of a compound that is known to be harmful to human health is high, the explained variability will indicate which sources are responsible for most of its emissions and what fraction of the total observed mass is contributed by each source. Therefore, this allows planning mitigation strategies.

Bootstrap runs were performed to ascertain the magnitude of random errors of the dataset (Norris et al., 2014; Paatero et al., 2014). Random errors can be caused due to the existence of infinite solutions with different $g_{ik}$, $f_{kj}$ and $e_{ij}$ matrices but identical $Q = \sum_{i=1}^{n}\sum_{j=1}^{m}(e_{ij}/\sigma_{ij})^2$. In the bootstrap runs, the timeseries is partitioned into smaller segments of a user specified length and the PMF is run on each of these smaller segments, for the same number of factors as the original model run. The model output of each bootstrap run is mapped onto the original solution using a cross correlation matrix of the factor contributions $g_{ik}$ of a given bootstrap run with the factor contributions $g_{ik}$ of the same time segment of the original solution using a threshold of the Pearson's correlation coefficient ($R$) > 0.6 as suggested by Norris et al. (2008, 2014). The bootstrap factor is assigned to the factor with which it is most strongly positively correlated, as long as the value of $R$ is greater than 0.6. If it cannot be attributed to any factor of the original solution it will be termed unmapped. The presence of a high fraction unmapped factor ($> 20\%$) is a clear indication of large random errors (introduced by a few critical observations that drastically impact factor profiles) and should be investigated carefully (Norris et al., 2014). In our analysis, no unmapped factors were present.

For each factor, the factor profile of all bootstrap runs combined is compared with the profile of the original model output. The model provides a box and whisker plot for the mass loading ($\mu g\,m^{-3}$) and percentage of each compound attributed to the factor profile of each of the factors during the bootstrap runs. It also ascertains for each compound whether or not the original solution for that factor falls into the interquartile range of the bootstrap results and provides this information in a table format.

When all sources are equally strong throughout the entire period, this bootstrap model provides a robust estimate of the total random error. However, if one of the sources is completely absent for a significant fraction of the total hours (like the brick kiln source throughout the first 13 days of the SusKat-ABC campaign), the bootstrap model may overestimate the random error substantially. For such a source, mass loading of all the compounds that contribute strongly to the factor profile of the source will typically be outside the interquartile range. For the same set of compounds, similar behavior could also be seen for the factor profile of several other factors. In such a situation, the error estimate of the bootstrap runs should only be considered as the upper limit of the potential random error.

In addition to the random error, the PMF model also has rotational ambiguity (Ulbrich et al., 2009; Paatero et al., 2014). This rotational ambiguity is caused due to the existence of multiple solutions which have a Q similar to the solution produced by the PMF model but different factor profiles and factor contributions. Thus, the model will find different local minima of the residual matrix, while determining the factor contribution matrix ($g_{ik}f_{kj}$). The coexistence of different solutions for the factor contribution matrix ($g_{ik}f_{kj}$) with the same sum of the scaled residuals $Q = \sum_{i=1}^{n}\sum_{j=1}^{m}(e_{ij}/\sigma_{ij})^2$ is called the rotational ambiguity of the model. The PMF 5.0 has a new feature named as "the constrained model operation" in which the rotational ambiguity of the model can be constrained using external knowledge of the source composition ($f_{kj}$) or contribution ($g_{ik}$) matrix. For instance, if a source was inactive for a particular period, then the contribution due to that factor during that time period could be pulled to zero in the model to provide more robust output. Alternatively, the emission ratios obtained from a particular source through samples collected at the source can also be used to constrain the model. Constraining the PMF model using such external knowledge gives rise to a penalty in $Q$ (the object function) and a maximum penalty of $5\%$ is recommended as a reasonable threshold (Paatero and Hopke, 2009). A detailed discussion of the use of constraints to a receptor model has been provided in previous studies (Norris et al., 2008, 2014; Paatero et al., 2002, 2014; Paatero and Hopke, 2009; Rizzo and Scheff , 2007).

## 2.5 Implementation of PMF

PMF was applied to the hourly averaged dataset of 37 ions measured using a Proton Transfer Reaction Time of Flight Mass Spectrometer (PTR-TOF-MS). All relevant analytical details pertaining to the site description, meteorology, sampling and quality assurance of the NMVOC dataset has already been described in detail in the companion paper to this special issue (Sarkar et al., 2016).

All the available data were used for the PMF analysis and the missing values were replaced by a missing value indicator (-999). To ensure that differential uncertainties do not drive the object function $Q$ and give undue weighting to calibrated organic ions while constructing source profiles, we followed the procedure used by Leuchner and Rappenglück (2010) for source apportionment of NMVOCs in the Houston Ship Channel area, assigning a constant uncertainty of $20\%$ for all the

ions. Due to its erratic timeseries profile, HCN ($m/z = 28.007$) was classified as a weak species in the PMF input while all other ions were classified as strong species. For weak species, the stated uncertainty is tripled, to reduce their impact on the scaled residual and hence Q. All the input data was converted from mixing ratios of ppb to mass concentrations ($\mu g\,m^{-3}$) using the relevant temperature, pressure and molecular weight and t he total measured NMVOC concentration was calculated by adding the mass concentrations of all measured NMVOCs. This conversion allows calculating the explained variability (Gaimoz et al., 2011) for the total VOC mass and comparing the results with emission inventories. The conversion does not introduce significant additional uncertainty and the variability induced by the temperature (average range observed was: 5-20°, C) has largely been taken into account by running the model with a 5% extra modelling uncertainty. The total VOC mass is classified as a weak species in the PMF input (Norris et al., 2014). All the measured ions had a signal to noise (S/N) ratio greater than 2. Table S2 of the supplementary information shows the signal to noise (S/N) ratios for all input NMVOC species used in the PMF along with other statistical parameters of the dataset.

PMF model runs ranging from 5 to 12-factor numbers were carried out to ascertain the best solution for this study, consistent with the chemical environment of the Kathmandu Valley. Based on the $Q/Q_{theoretical}$ ratio, the physical plausibility of the factors and constraints imposed by the rotational ambiguity of the solution, an 8-factor solution was deemed to be the best for this dataset. For the data presented in this study, the $Q/Q_{theoretical}$ ratio is <1 even for a 3 factor solution with no physical plausibility and and hence the absolute number does not help to decide the optimum number of factors. Supplementary Figure S2 shows clearly, that the number of factors has almost no impact on how well the total mass is reproduced by the model, but the last distinct drop in the $Q/Q_{theoretical}$ ratio is seen when the number of factors is increased to 8. When fewer than 7-factors were employed, several source profiles appeared to be mixed (Figure S3a,b), indicating inadequate resolution of sources. The solution incorporating 7-factors was considered inappropriate, as the daytime biogenic emissions and photochemical sources could not be separated from the nighttime combustion source of isoprene in the 7-factor solution. Even when the model was nudged towards separating the biogenic emissions and the anthropogenic combustion sources of isoprene using the constraint mode, this separation could only be accomplished with a large penalty on Q in the 7-factor solution. The 9-factor solution had too much rotational ambiguity and assigned brick kiln emissions to two largely co-linear factors, both of which had an incomplete source profile with respect to aromatic compounds and were essentially created to better account for minor variations in the emission ratios associated with brick kiln emissions during the firing up period and the continuous operation later in the campaign (Figure S3c).

The diagnostics for the 8-factor solution are summarized in Table 1. The eight factors were - 1) traffic, 2) residential biofuel use and waste disposal, 3) mixed industrial emissions, 4) biomass co-fired brick kilns, 5) unresolved industrial emissions, 6) solvent evaporation, 7) mixed daytime

source and 8) biogenic emissions. A detailed description for the identification and the attribution of the 8-factor solutions is provided later in section 3.1. The primary data strongly supports an 8 factor solution. The top 2-3 compounds explained by each of the 8 factors have a much higher R when their input time series is correlated compared to the R obtained when their time series is correlated with the time series of any other compound (Supplementary Table S5).

The traffic factor explains more than 60 % of the variability of Toluene, C-8 and C9 aromatics. The time series of Toluene, C8 and C9 aromatics correlates with R >0.96 for all possible pairs when the original time series of these compounds are correlated with each other. The R of the time series of these same compounds with the time series of styrene is lower 0.81-0.85 while a correlation of their time series with all other compounds yields R <0.78. This indicates toluene, sum of C-8 and C9 aromatics share a major common source with each other which is not shared by other compounds, namely the traffic source. Hence a less than 6 factor PMF solution which is incapable of capturing the traffic source is not a better representation of the reality.

For styrene the highest correlation is with furan R=0.87 indicating that the two compounds have a significant source in common, which styrene also shares with higher aromatics and propyne (R=0.86), but the lower R of styrene with the aromatic compounds indicates that styrene has at least two dominant sources with distinct emission ratios. These sources are the traffic source (explaining roughly 40 % of the styrene) and the residential burning source which explains 30 % of the styrene and furan variability. These two sources are separated only with a 6 factor solution.

Benzene has a strong source in the form of biomass co-fired brick kilns which results in a distinct increase in emission at the time the brick kilns restart their operations. This source is shared with acetonitrile (R=0.89), nitromethane (R=0.82) and naphthalene (R=0.81) but all of these compounds also have other sources which are either not shared with benzene or have different emission ratios. This source appears in the 3 factor solution but its source profile is contaminated with mixed industrial emission. The closure period of brick kilns is only fully captured and restricted to the brick kiln factor after the number of factors is increased to 7.

The mixed industrial source explains 66 % of the ethanol variability, but this compound has a relatively low R with all other compounds (0.73 with propene and 0.7 with nitromethane and acetonitrile <0.66 with the rest) indicating that there must be at least two distinct ethanol sources with different source fingerprints. A second distinct ethanol source in the form of solvent evaporation, however, separates from the mixed daytime factor only in the 7 factor solution.

The mixed daytime factor primarily contains photo-chemically formed compounds most notably isocyanic acid, which shows a strong correlation with its own precursors formamide (R=0.85) and acetamide (R=0.82). Figure S8 presents reaction schematic for the formation of formamide and isocyanic acid. This compound has a much weaker correlation with other compounds, which have other sources in addition to the photochemical source (R=0.5 to 0.58 for formaldehyde, acetaldehyde, the nitronium ion, formic acid and acetic acid). This factor should ideally be restricted to photo-

**Table 1.** Diagnostic for the results of the positive matrix factorization (PMF) model run

| | |
|---|---|
| n (samples) | 1006 |
| m (species) | 37 |
| k (factors) | 8 |
| Q (theoretical) | 4480.37 |
| Q (model) | 4562.89 |
| Mean ratio NMVOC(estimated)/NMVOC(observed) | 0.999 |

chemically formed secondary compounds, however, it remains heavily contaminate with night-time primary emissions during the second half of the campaign till the number of factors is increased to 8 (Figure S3c). Even the 8 and 9 Factor solution still contain some minor contamination from primary emissions. Hence the name of the source is retained as mixed daytime source.

The solvent evaporation factor is characterised by acetaldehyde and acetic acid which have their strongest correlation with each other (R=0.82). Apart from this, the defining compound, acetaldehyde, shows moderate correlation with formaldehyde (R=0.72) and acetone (R=0.68) but only the former correlates with acetic acid (R=0.85) as it shares both the solvent evaporation source and the photo-oxidation source with acetaldehyde, while the later (acetone) correlates much stronger with methyl ethyl ketone (R=0.95) and methyl vinyl ketone (R=0.86) and isoprene (R=0.79) and hence shares the biogenic emission source in addition to the the solvent evaporation factor. While these three daytime sources are resolved in the 7 factor solution their source profiles continue to be contaminated with primary emissions. While the same can be pushed around from the biogenic factor into the mixed daytime factor using rotational tools, they cannot be sufficiently removed from both till an 8th factor is allowed.

The unresolved industrial emission factor explains a significant fraction of the 1,3-butadiyne which shares most of its sources with methanol (R=0.9). The source profile also captures several other compounds with a lower correlation with 1,3-butadiyne including propanenitrile (R=0.86), acrolein + methylketene (R=0.82) and propene (R=0.8). The R obtained while cross correlating the time series of 1,3-butadiyne with that of ethanol, the defining compound of the mixed industrial source profile, is only 0.73 and ethanol correlates only weakly with Acrolein + Methylketene (R=0.59) indicating that these mixed industrial emissions and unresolved industrial emissions represent distinct sources, which can only be resolved in a 8 factor solution.

To identify the uncertainty associated with the PMF solution, bootstrap runs were performed 100 times taking 96 hours as the segment length. This is slightly shorter than the recommended length based on the equation of Politis and White (2004), of 108 hours but represents a multiple of 24 hours and hence ensures each bootstrap run contains four full days' worth of data. There were no unmapped factors in the bootstrap runs.

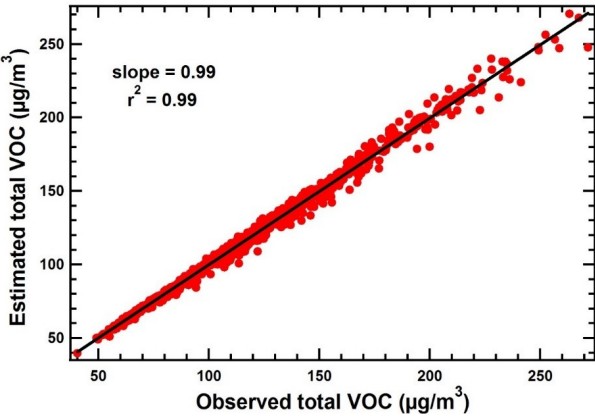

**Figure 2.** Correlation between estimated and observed NMVOC concentrations

Figure 2 shows the correlation between the estimated total measured NMVOC concentrations calculated using the contributions from all factors (vertical axis) with measured total measured NMVOC concentrations (horizontal axis). An excellent correlation ($r^2 = 0.99$) indicates that PMF model can explain almost all variance in the total measured NMVOC concentrations.

The constrained model mode was used to further improve the 8-factor solution. The constraint mode is a new rotational tool introduced in the 5.0 version of the EPA PMF as an alternative to the FPeak module. The constraint mode allows to exploit the rotational ambiguity of the model to push the PMF solution into a physically more realistic space. It uses pre-existing knowledge such as source fingerprints, source emission ratios or activity data. We found that when the two modules were compared for an equal number of factors the constraint mode performance was superior to the FPeak module. The original model output showed positive correlations between the factor contribution time series of the biomass co-fired brick kilns and mixed industrial emissions ($r^2 = 0.27$) factors as well as the residential biofuel use and waste disposal factor with traffic factor ($r^2 = 0.42$). Since this is a new feature and has only recently been used by Brown et al. (2015) for ambient air data, a detailed description of the implementation procedure and an analysis of how the constraints affected the model output is provided here. Several constraints were used to obtain a more robust PMF solution.

First, the upper limit for the emission ratio of the individual aromatic compounds to isoprene as reported by Misztal et al. (2015) were used to constrain the factor profile of primary biogenic emissions. As a small fraction of the biogenic isoprene gets attributed to other daytime factor (mixed daytime) by the PMF model, the same constraints were used on mixed daytime factor and the solvent evaporation factor as well.

Second, it was assumed that aromatic compounds and acetonitrile are not photochemically produced. Acetic acid is associated with both mixed daytime and solvent evaporation, and so the ratios

of aromatic compounds and acetonitrile to acetic acid were nudged towards 0.0001 for these two factors.

Third, to improve the representation of brick kiln emissions, and the residential biofuel use and waste disposal in the model, the respective factors, which were clearly identified in the original model solution, were nudged using the emission ratios of aromatic compounds to benzene from

grab samples of domestic waste burning (garbage burning grab sample) and fixed chinmney bull's trench brick kiln emissions (FCBTBK grab sample) collected directly at the point source. This was required, because in the original model output, the residential biofuel use and waste disposal factor correlated with the traffic factor ($r^2$ = 0.42) while the brick kilns emission factor correlated with the mixed industrial emissions factor ($r^2$ = 0.27). This indicates that there was substantial rotational

ambiguity for these two factor pairs.

Nudging was performed by exerting a soft pull allowing for a maximum 0.2 % change in $Q$ for each constraint. A soft pull allows the change in the $Q$ value up to a certain limit by pulling the values to a target value for an expression of elements (the emission ratio). If no minima can be found for which the change in $Q = \sum_{i=1}^{n}\sum_{j=1}^{m}(e_{ij}/\sigma_{ij})^2$ is less than 0.2 % in the $g_{ik}f_{kj}$ matrix after $f_{kj}$

has been constrained, no change was made and the original solution was retained. If the condition can be met without changing $Q$ by more than the threshold, the revised factor profiles will be used as the base upon which the next constraint in the list of constraints will be executed.

Implementing the constraints mentioned above, significantly improved the representation of biogenic emissions, mixed daytime and solvent evaporation factors. Figure S4 of the supplementary

information shows a comparison of the box and whisker plots of the biogenic emissions, mixed daytime and solvent evaporation factors before and after nudging and demonstrates the significant improvement after applying constraints.

After nudging, the contribution of the biogenic factor correlated better with solar radiation ($r^2$ = 0.48) while the mixed daytime factor correlated better with ambient temperature ($r^2$ = 0.42).

The factor profile of the solvent evaporation correlates better with the rise in solar radiation and temperature after sunrise (07:00 - 09:00 LT; $r^2$ = 0.53). Table 2 represents the emission ratios used to nudge the biogenic, mixed daytime and solvent evaporation factors and provides the corresponding emission ratios (ERs) before and after nudging.

It can be seen that most constraints on the aromatic to isoprene ratio could be executed without

exceeding the penalty on $Q$. In the biogenic factor, only the naphthalene/isoprene ratio could not be constrained. The solvent evaporation and mixed daytime factors contain only a small fraction of the total daytime isoprene (8 % and 7 %, respectively). Given the very small overall isoprene mass in these two factor profiles, few additional ratios did not meet the constraining criteria in these factor profiles (namely acetonitrile/isoprene and trimethylbenzenes/isoprene ratio in the mixed daytime

factor and the xylenes/isoprene and naphthalene/isoprene ratio in the solvent evaporation factor).

**Table 2.** Inter NMVOC emission ratios used for biogenic, solvent evaporation and mixed daytime factors to nudge the PMF model and the corresponding emission ratios before and after nudging

| ERs/Isoprene | ERs used to nudge | BG before nudging | after nudging | SE before nudging | after nudging | MD before nudging | after nudging |
|---|---|---|---|---|---|---|---|
| Acetonitrile | 0.002 | 0.06 | 0.00 | 0.00 | 0.004 | 2.78 | 1.75 |
| Benzene | 0.002 | 0.29 | 0.00 | 0.52 | 0.00 | 0.15 | 0.00 |
| Toluene | 0.012 | 0.10 | 0.01 | 0.39 | 0.00 | 4.82 | 0.00 |
| Styrene | 0.002 | 0.02 | 0.00 | 0.06 | 0.00 | 0.00 | 0.002 |
| Xylenes | 0.002 | 0.00 | 0.0002 | 0.35 | 0.41 | 4.65 | 0.00 |
| Trimetylbenzenes | 0.002 | 0.06 | 0.01 | 0.09 | 0.00 | 1.85 | 0.20 |
| Naphthalene | 0.002 | 0.31 | 0.30 | 0.36 | 0.60 | 0.00 | 0.002 |
| ERs/Acetic acid | ERs used to nudge | BG before nudging | after nudging | SE before nudging | after nudging | MD before nudging | after nudging |
| Acetonitrile | 0.0001 | 0.57 | 0.00 | 0.00 | 0.0001 | 0.07 | 0.09 |
| Benzene | 0.002 | 1.48 | 0.00 | 0.04 | 0.00 | 0.01 | 0.00 |
| Toluene | 0.0001 | 1.01 | 0.004 | 0.05 | 0.00 | 0.12 | 0.00 |
| Styrene | 0.0001 | 0.15 | 0.00 | 0.01 | 0.00 | 0.00 | 0.0001 |
| Xylenes | 0.0001 | 0.00 | 0.0001 | 0.04 | 0.01 | 0.12 | 0.00 |
| Trimetylbenzenes | 0.0001 | 0.59 | 0.004 | 0.01 | 0.00 | 0.05 | 0.01 |
| Naphthalene | 0.0001 | 3.08 | 0.15 | 0.04 | 0.01 | 0.00 | 0.0001 |

BG = Biogenic; SE = Solvent evaporation; MD = Mixed daytime

Some of these compounds (such as naphthalene) could not be constrained in the same factors while constraining the ERs with respect to acetic acid.

The fact that the constrained run was incapable of removing naphthalene from the source profiles of the biogenic and the solvent evaporation source and the fact that the diel profiles of both these factors show a weak secondary peak between 17:00 - 22:00 LT, seems to indicate that an additional weak combustion source with a high naphthalene emission ratio is possibly poorly represented by the current 8-factor solution. Cooking on 3-stone fires is known to emit large amounts of benzene and naphthalene (Stockwell et al., 2015) and the temporal profile of such a cooking source could overlap with that of the garbage fires. It can be noted that 3-stone fires is still a common way to cook for construction workers and brick kiln workers staying in temporary camps in the Kathmandu Valley. This would make it challenging for the model to separate these two sources. We will henceforth refer to the garbage burning factor as the residential biofuel use and waste disposal factor.

**Table 3.** Comparison of aromatics/benzene ERs (emission ratios) obtained from PMF (before and after nudging), respective grab samples, the 3-stone firewood source reported in Stockwell et al. (2015) and the mixed garbage burning and open cooking fire sources reported in Stockwell et al. (2016)

| ERs/Benzene | FCBTBK grab samples | BK PMF (before nudging) | BK PMF (after nudging) | garbage burning grab samples | RB+WD PMF (before nudging) | RB+WD PMF (after nudging) | 3-stone firewood[1] | Mixed garbage[2] | Open hardwood cooking[2] |
|---|---|---|---|---|---|---|---|---|---|
| Toluene | 0.80 | 0.28 | 0.35 | 0.34 | 0.33 | 0.34 | 0.11 | 0.37 | 0.27 |
| Styrene | 0.08 | 0.05 | 0.06 | 0.16 | 0.22 | 0.18 | 0.09 | 0.19 | 0.11 |
| Xylenes | 0.58 | 0.16 | 0.22 | 0.25 | 0.28 | 0.25 | 0.10 | 0.18 | 0.12 |
| Trimethylbenzenes | 0.31 | 0.06 | 0.09 | 0.08 | 0.16 | 0.12 | 0.03 | 0.02 | 0.03 |
| Naphthalene | 0.09 | 0.14 | 0.15 | 0.09 | 0.16 | 0.11 | 0.40 | - | - |

1. Stockwell et al. (2015); 2. Stockwell et al. (2016); BK = Biomass co-fired brick kilns; RB+WD = Residential biofuel use and waste disposal

Figure S5a of the supplementary information shows the G-space plots for two factors, namely biomass co-fired brick kilns and mixed industrial emissions. A stronger correlation ($r^2 = 0.42$) existed in the original solution prior to nudging with ERs of FCBTBK grab samples, which reduced to $r^2 = 0.18$. Similarly, after nudging with ERs of the garbage burning grab sample the correlations between residential biofuel use and waste disposal was reduced from 0.27 to 0.18, as shown in Figure S5b. Thus, the new solution fills the solution space better.

Table 3 summarizes the aromatics/benzene emission ratios derived from the PMF (before and after nudging) and its comparison with the emission ratios obtained from grab samples for biomass co-fired brick kilns and residential biofuel use and waste disposal sources. These emission ratios are also compared with the ERs reported for 3-stone firewood stoves in Stockwell et al. (2015) and the mixed garbage burning and open cooking fire sources reported for Nepal in Stockwell et al. (2016).

For the residential biofuel use and waste disposal source, the original model run already had emission ratios very similar to the garbage burning grab samples of the garbage burning fire. The constrained run improved the agreement further for styrene, trimethylbenzenes and naphthalene. Constraining this factor with the ERs of 3-stone firewood stoves from Stockwell et al. (2015) instead of our garbage burning grab samples resulted in a larger penalty on $Q$ and did not improve the representation of the biogenic, mixed daytime and solvent evaporation factors.

For brick kilns, the emission ratios of the constrained model output runs diverged from the emission ratios of the FCBTBK grab samples. However, the temporal profile of the activity, especially the closure of the brick kilns during the first part of the campaign is better captured by the constrained run and the correlation with mixed industrial emission sources reduced significantly. The FCBTBK grab samples were collected on 6 December 2014, two years after the SusKat study, so differences from the emission profiles observed during the SusKat-ABC campaign are a possibil-

ity. Alternatively, the differences could also stem from the inherently variable nature of this source. In particular, naphthalene and benzene were higher in the source profiles of the SusKat-ABC campaign compared to their relative abundances in the FCBTBK grab samples. At the time the FCBTBK grab samples were collected (on 6 December 2014), brick kilns were co-fired using coal, wood dust and sugarcane extracts. It is possible that in January, during peak winter season, a different type of biomass, one associated with higher benzene and naphthalene emissions (e.g. wood) was used in these biomass co-fired brick kilns, resulting in the slight disagreement between the PMF source profile and FCBTBK grab sample signature for this factor. Table S3 of the supplementary information shows the percentage contribution of PMF derived factors obtained from constrained runs with 5, 6, 7, 8 and 9-Factors.

## 2.6 Conditional probability function (CPF) analyses

For identifying the physical locations associated with different local sources, conditional probability function (CPF) analyses were performed. CPF is a well-established method to identify source locations of local sources based on the measured wind (Fleming et al., 2012). In CPF, the probability of a particular source contribution from a specific wind direction bin exceeding a certain threshold is employed which is calculated as follows:

$$CPF = \frac{m_{\Delta\theta}}{n_{\Delta\theta}} \tag{4}$$

Where $m_{\Delta\theta}$ represents the number of data points in the wind direction bin $\Delta\theta$ which exceeded the threshold criterion and $n_{\Delta\theta}$ represents the total number of data points from the same wind direction bin. For this study, $\Delta\theta$ was chosen as $30°$ and data for wind speed $> 0.5\,\mathrm{m}^{-1}$ were used.

## 2.7 Calculation of ozone and SOA formation potential

The ozone formation potential of individual NMVOCs was calculated as described by the following equation (Sinha et al., 2012):

$$Ozone\ production\ potential = \left(\sum k_{(VOC_i+OH)}[VOC]_i\right) \times OH \times n \tag{5}$$

For the ozone production potential calculation, the average hydroxyl radical concentration was assumed to be $[OH] = 1{\times}10^6\ \mathrm{molecules\,cm}^{-3}$ with n = 2 and only data pertaining to the mid-daytime period were considered (11:00 - 14:00 LT).

SOA yield of a particular NMVOC depends on the NOx conditions and Pudasainee et al. (2006) previously reported NOx-rich conditions in the Kathmandu valley. Therefore, SOA production was calculated by using reported SOA yield at high NOx conditions according to the following equation:

$$SOA\ production = [VOC]_i \times SOA\ yield\ of\ VOC_i \tag{6}$$

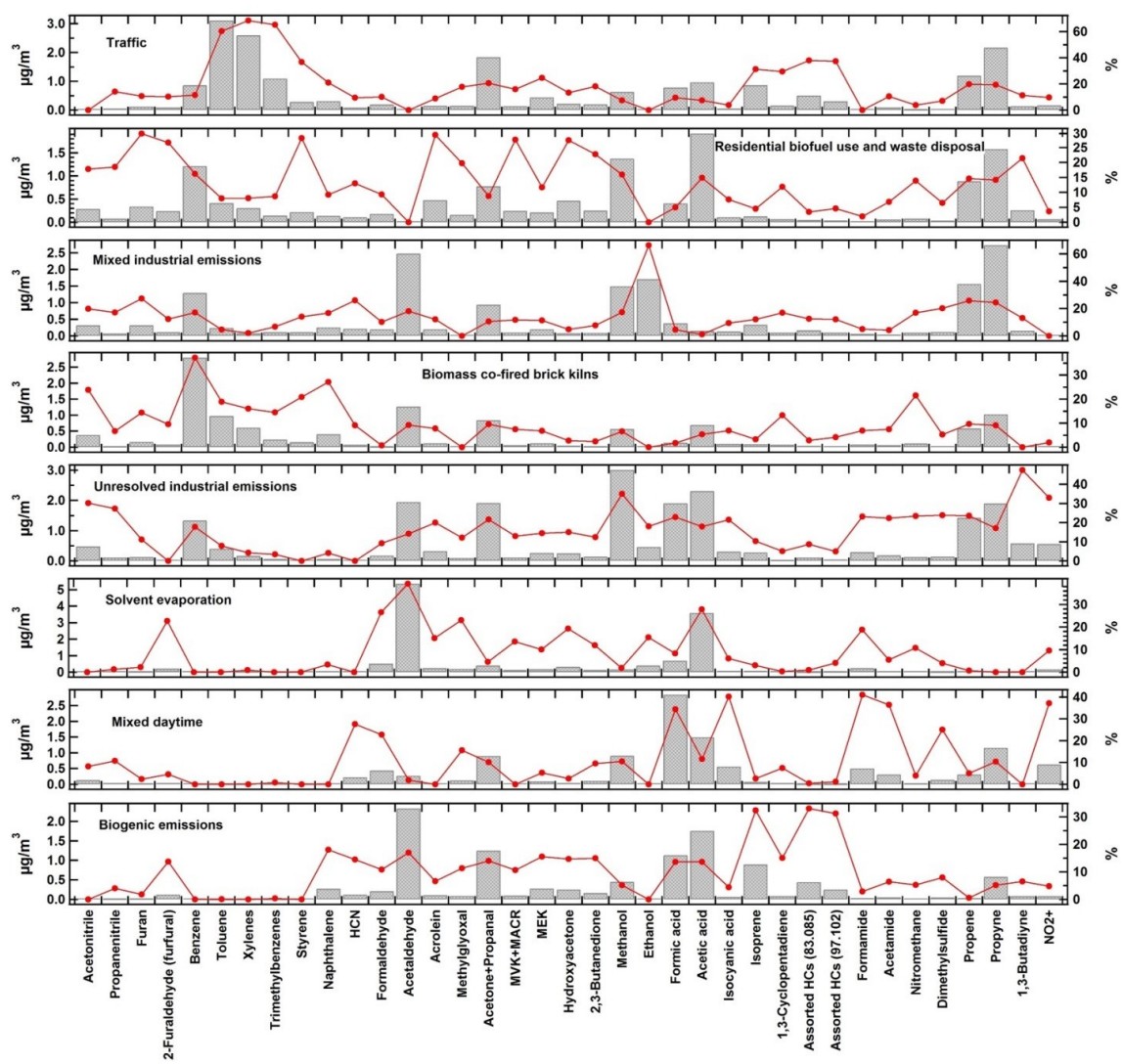

**Figure 3.** Factor profiles of the eight sources obtained by PMF analysis

## 3 Results and Discussion

### 3.1 Identification of PMF Factors

Figure 3 represents the factor profiles of all the eight factors resolved by the PMF model in which grey bars (left axis) indicate the mass concentrations and red lines with markers (right axis) show the percentage of a species in the respective factor.

Identification and attribution of these factors is discussed in detail in the following sections.

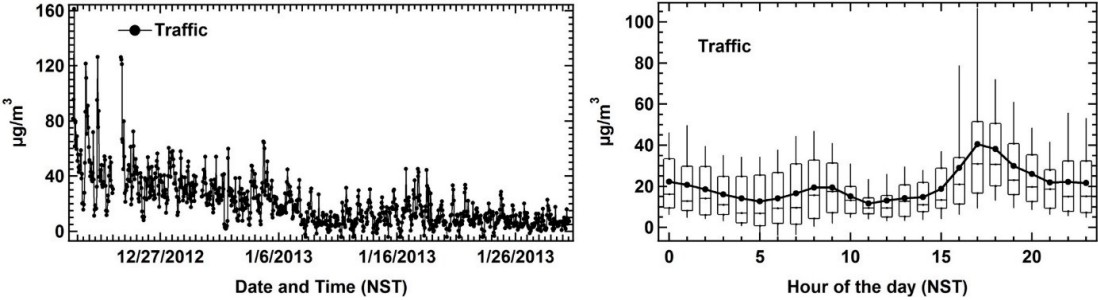

**Figure 4.** Timeseries and diel box and whisker plot for Factor 1 (Traffic)

### 3.1.1 Factor 1 - Traffic

More than 60 % of the total toluene, sum of C8-aromatics, sum of C9-aromatics and $\sim 37$ % of the
total assorted hydrocarbons ($m/z =$ 97.102 and 83.085) were explained by Factor 1. Toluene and
C8-aromatics contributed most ($\sim 16$ % and $\sim 13$ %, respectively) to the total measured NMVOC
mass of Factor 1. In addition four other compounds also contributed $\geq 5$ % to the total mass of this
factor (propyne ($\sim 11$ %), acetone ($\sim 9$ %), propene ($\sim 6$ %) and sum of C9-aromatics ($\sim 5$ %)).
The other 31 NMVOCs contributed $\sim 40$ % of the total measured NMVOC mass to this factor but
their individual contributions were $\leq 5$ % each. The diel profile of Factor 1 (Figure 4) showed char-
acteristic evening peak at 17:00 LT with an average concentration of $\sim 40\,\mu\mathrm{g\,m}^{-3}$. This evening
peak showed large variability and plume-like characteristics as the average and median diverged
frequently. Occasionally, the mass contribution of this factor amounted to $\sim 100$ $\mu\mathrm{g\,m}^{-3}$. The high
variability during the evening peak hour indicates that the source strength is not equal for all wind
directions, but varies with fetch region.

Table 4 shows that the aromatics/benzene emission ratios for this factor are in good agreement
with the emission ratios reported by previous studies for vehicular emissions in tunnel experiments
and in metropolitan sites/megacities. In view of the diel profile and observed chemical signatures,
Factor 1 was attributed to traffic. It can be noted that in winter, rush-hour in the city starts at 16:00
LT, while westerly winds still bring urban air to the measurement site. The morning rush hour
in the city takes place in calmer winds which leads to a less sharp peak. It is interesting to note
that $\sim 37$ % of the total styrene was present in this factor and $\sim 31$ % of the total isoprene was
also explained by this factor. Few previous studies employing GC-FID have reported traffic related
sources of isoprene in urban areas (Borbon et al., 2001; Hellèn et al., 2012) and also estimated
isoprene as one of the top 10 contributors to OH reactivity from traffic (Nakashima et al., 2010). A
recent study suggested m/z 69 $\mathrm{C_5H_8H^+}$ could also result from the fragmentation of cycloalkanes
and cycloalkenes (Gueneron et al., 2015). Fragmentation of these compounds should also result in
product ions at m/z 111 and/or m/z 125 and the signal at those masses at  135 Td should be above
200 ppt considering the measured $\mathrm{C_5H_8H^+}$ ion signal in the Kathmandu valley during our study.

**Table 4.** Emission ratios of NMVOCs/benzene for aromatic hydrocarbons derived from the PMF model for factor attributed to traffic and comparison of ERs with previous studies for traffic source profiles

| ERs/Benzene | Kathmandu PMF | Tunnel study, Stockholm[1] | Tunnel study, Hong Kong[2] | Tunnel study, Taipei[3] | Mexico City[4] | Los Angeles[5] |
| --- | --- | --- | --- | --- | --- | --- |
| Toluene | 3.41 | 3.89 | 2.27 | 2.38 | 3.47 | 2.45 |
| C8-aromatics | 2.89 | 2.81 | 0.87 | 1.86 | 3.55 | 1.38 |
| C9-aromatics | 1.20 | - | 0.77 | 1.36 | 2.31 | 0.48 |
| Styrene | 0.30 | - | - | 0.39 | 0.17 | - |
| Naphthalene | 0.19 | - | 0.10 | - | - | - |

1. Kristensson et al. (2004); 2. Ho et al. (2009); 3. Hwa et al. (2002) ; 4. Bon et al. (2011) ; 5. Borbon et al. (2013)

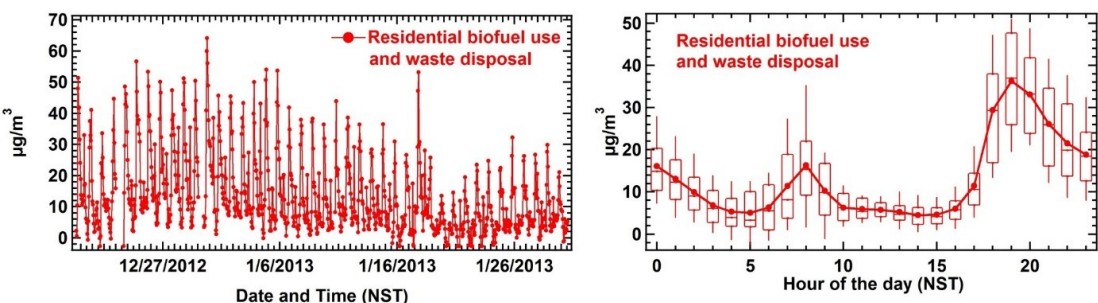

**Figure 5.** Timeseries and diel box and whisker plot for Factor 2 (Residential biofuel use and waste disposal)

However, in the observed mass spectra, there was no significant signal at these m/z values. Therefore, we conclude that isoprene is the more plausible assignment.

### 3.1.2 Factor 2 - Residential biofuel use and waste disposal

Factor 2, too, showed regular evening hour peaks and a bimodal profile (Figure 5). However, the evening peak of average concentrations as high as $\sim 40\,\mu\mathrm{g\,m^{-3}}$ occurred after the traffic peak (at 19:00 LT) and had less variability, indicating that this source is an area source that is spatially spread throughout the Katmandu Valley. The diel box and whisker plot also has a relatively weak morning peak (at 08:00 LT) with average concentrations of $\sim 18\,\mu\mathrm{g\,m^{-3}}$. Figure 3 shows, that this factor explains 30 % of the total styrene, furan, 2-furaldehyde and acrolein.

Most of the measured NMVOC mass in this factor was contributed by acetic acid, propyne, methanol, benzene, propene and acetone + propanal ($\sim 14\,\%$, $\sim 12\,\%$, $\sim 10\,\%$, $\sim 9\,\%$, $\sim 7\,\%$ and $\sim 6\,\%$ respectively). The other 31 measured NMVOCs contributed $\sim 42\,\%$ to this factor, but their individual contributions were $\leq 5\,\%$ each (Figure 3). It was observed that garbage/trash burning activities were more intense during evening hours in winter in the Kathmandu Valley. Table 5 shows a comparison of the aromatics/benzene emission ratios obtained from the PMF, with previously re-

**Table 5.** Emission ratios of NMVOCs/benzene for acetonitrile and aromatic hydrocarbons derived from the PMF model for the factor attributed to Residential biofuel use and burning household waste and comparison with previously reported studies and the garbage burning grab samples collected at the point source

| ERs/Benzene | Kathmandu PMF | Kathmandu garbage burning grab samples | Mixed garbage burning[1] | Household waste burning[2] | Open hardwood cooking[1] | Trash burning[3] | Scrap tires burning[2] |
|---|---|---|---|---|---|---|---|
| Acetonitrile | 0.23 | 0.77 | - | - | - | 0.06 | - |
| Toluene | 0.34 | 0.34 | 0.37 | 0.38 | 0.27 | 0.41 | 0.63 |
| C8-aromatics | 0.25 | 0.25 | 0.19 | 0.22 | 0.11 | 0.10 | 0.43 |
| C9-aromatics | 0.12 | 0.08 | 0.18 | - | 0.12 | 0.03 | 0.03 |
| Styrene | 0.18 | 0.16 | 0.02 | 0.54 | 0.03 | 0.86 | 0.30 |
| Naphthalene | 0.11 | 0.09 | - | 0.01 | - | 0.10 | 0.30 |

1.Stockwell et al. (2016) ; 2. Lemieux et al. (2004) ; 3. Stockwell et al. (2015)

ported aromatics/benzene ratios for waste and trash burning, and with the emission ratios of garbage burning grab samples that were collected in the Kathmandu Valley near the point source (a household waste fire). It can be seen that the aromatics/benzene emission ratios of the PMF output are in excellent agreement with the values obtained for garbage burning grab samples collected in the Kathmandu Valley.

There is some agreement with the emission ratios reported in previous studies, though all of these previous studies found higher emission ratios for styrene. This could indicate that the composition of household waste in the Kathmandu Valley is different (less polystyrene, plastic and more biomass) or that the source profile is mixed with that of a second source, with similar spatial and temporal characteristics. Residential biofuel use is expected to have a similar temporal profile and did not
appear as a separate factor in the PMF solution. Therefore, Factor 2 was attributed to residential biofuel use and waste disposal sources collectively.

### 3.1.3   Factor 3 - Mixed industrial emissions

This factor explained 66 % of the total ethanol, which is used as an industrial solvent. Moreover, $\sim 20 - 25$ % of the total propyne, propene, acetonitrile, dimethyl sulfide (DMS) and furan were also
present in this factor. All these compounds have industrial sources (Karl et al., 2003; Kim et al., 2008) as they are widely used as solvents/reactants in various industrial processes and can be emitted during combustion processes. Therefore, Factor 3 was attributed to mixed industrial emissions. Most of the measured NMVOC mass in this factor was contributed by propyne ($\sim 16$ %), acetaldehyde ($\sim 15$ %), ethanol ($\sim 10$ %), propene ($\sim 9$ %), methanol ($\sim 9$ %), benzene ($\sim 8$ %) and acetone + propanal ($\sim$
5 %). The emissions reflect both release of chemicals used in the industrial units as well as emissions

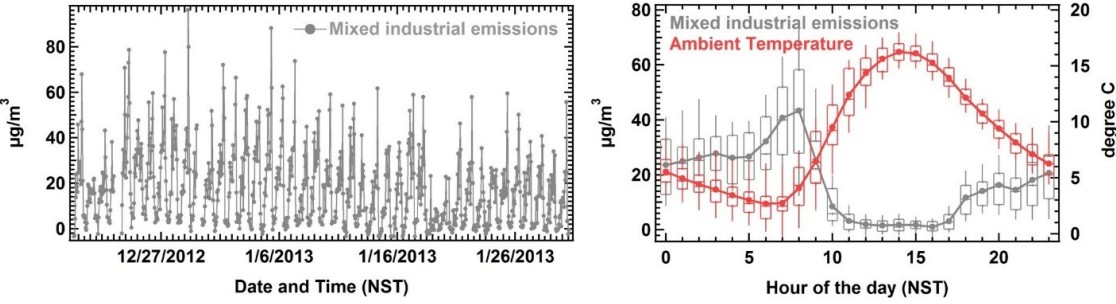

**Figure 6.** Timeseries and diel box and whisker plot for Factor 3 (Mixed industrial emissions)

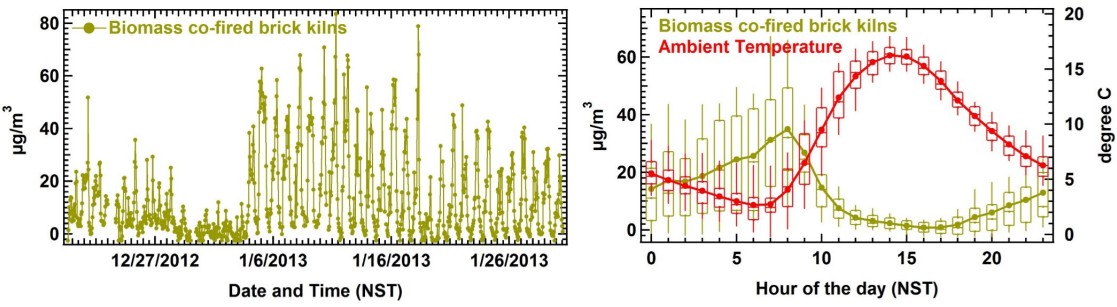

**Figure 7.** Timeseries and diel box and whisker plot for Factor 4 (Biomass co-fired brick kilns)

associated with combustion of a variety of fuels including biofuels. The other 30 NMVOCs jointly contributed only $\sim 28\,\%$ of the total measured NMVOC mass and their individual contribution were $\leq 5\,\%$ each. The emission strength of industrial sources is typically constant throughout the day and hence the observed mass concentrations are driven by boundary layer dynamics. The diel box and whisker plot (Figure 6) shows a gradual increase in the mass concentrations throughout the night. The highest mass concentration are observed just after sunrise, when the inversion in the mountain Valley is most shallow. This shallow early morning boundary layer is caused by the cold pooling of air at night, which results in an accumulation of cold air at the Valley bottom. The rising sun first warms the upper part of the Valley's atmosphere, while the Valley bottom is still in the shade of the surrounding mountains. Once direct sunlight reaches the Valley bottom, warming and thermally driven convection breaks the shallow boundary layer and wind speeds increase, increasing turbulent mixing under a growing boundary layer. The daytime mass concentrations of the mixed industrial emissions are hence an inverse of the temperature and wind speed profile (Figure 6).

### 3.1.4 Factor 4 - Biomass co-fired brick kilns

The diel box and whisker plot of factor 4 (Figure 7) shows a profile that is similar to the profile of mixed industrial emissions, indicating that this factor should be attributed to a source that operates 24/7, as its mass loadings, too, represent an inverse of the temperature and wind speed profile. The

timeseries of Factor 4 showed sudden increase from 4 January 2013 at exactly the time when brick kilns in the Kathmandu Valley became operational (Sarkar et al., 2016).

Benzene ($\sim 23\,\%$) contributed most to the total measured NMVOC mass of Factor 4. In addition acetaldehyde ($\sim 10\,\%$), propyne ($\sim 8\,\%$), toluene ($\sim 8\,\%$), acetone ($\sim 7\,\%$), acetic acid ($\sim 5\,\%$) and xylenes ($\sim 5\,\%$) also contributed significantly to the total measured NMVOC mass. The other 30 NMVOCs contributed $\sim 34\,\%$ to the total measured NMVOC mass of this factor, but their individual contribution were $\leq 5\,\%$ each. Overall, factor 4 explained $\sim 37\,\%$ of the total benzene and $\sim 24\,\%$

of the total acetonitrile mass loading.

It is reported that brick kilns in the Kathmandu Valley burn large quantity of biomass, wood and crop residues along with coal (Stone et al., 2010; Sarkar et al., 2016) that can lead to significant emission of aromatics and acetonitrile (Akagi et al., 2011; Yokelson et al., 2013; Sarkar et al., 2013). Therefore Factor 4 was attributed to the biomass co-fired brick kilns and the conditional probability

function analysis (section 3.2) is consistent with this assignment.

### 3.1.5    Factor 5 - Unresolved industrial emissions

Factor 5 explained $\sim 48\,\%$ of the total 1,3-butadiyne, $\sim 35\,\%$ of the total methanol, $\sim 30\,\%$ of the total acetonitrile and $27\,\%$ of the total propanenitrile and $24\,\%$ of the total nitromethane. 1,3-butadiyne is used in the production of several polymers and acetonitrile and propene can be side products in

this process. Propanenitrile is used to start acrylic polymerization reactions in industrial processes. The largest use of methanol worldwide is as feedstock for the plastic industry and nitromethane is used in the synthesis of several important pharmaceutical drugs. It can be noted that several pharmaceutical industries are located in the Thimi area which is only $\sim 2\,km$ away from the measurement site. Nitromethane is also emitted from combustion of diesel fired generators (Inomata et al., 2013,

2014; Sekimoto et al., 2013) which are used as a back-up power source by both small and large industrial units in the Kathmandu Valley. It is, therefore, likely that miscellaneous nearby industries contributed significantly to the unresolved factor. The diel profile of Factor 5 (Figure 8) showed morning and evening peaks (at 09:00 - 10:00 LT and 17:00 LT, respectively), which is not typical for industrial emissions, but this factor always had a high background with average mass loadings

of $\sim 20\,\mu g\,m^{-3}$ throughout. The timeseries and diel profile (Figure 8) of this factor did not reveal characteristics that could be related uniquely to a known emission source.

Figure 8 displayed elevated daytime mass concentrations and an evening peak for this factor that occurs slightly before the traffic peak in the early evening during the first part of the SusKat-ABC campaign (until 25 December). Towards the end of the campaign (from 10 January onwards), the

same factor had diurnal variations that showed some similarity to profiles of both the solvent evaporation (morning peak) and mixed industrial emissions (slow rise throughout evening and nighttime) factors. Between 25 December and 10 January, diurnal patterns are weak and peaks in the unresolved factor seem to coincide with peaks in the solvent evaporation factor. This comparison of the

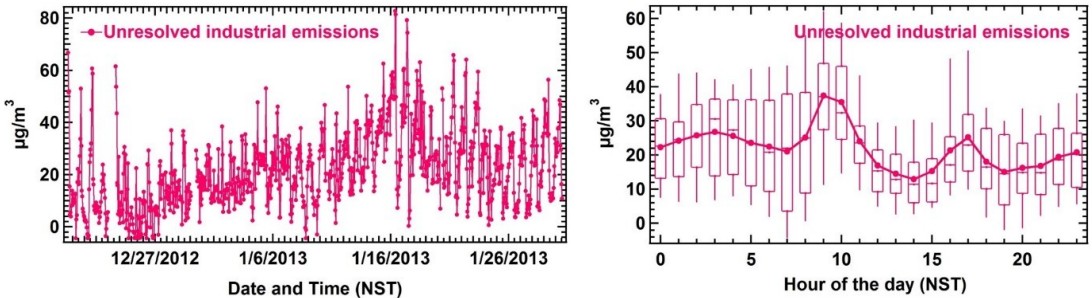

**Figure 8.** Timeseries and diel box and whisker plot for Factor 5 (Unresolved industrial emissions)

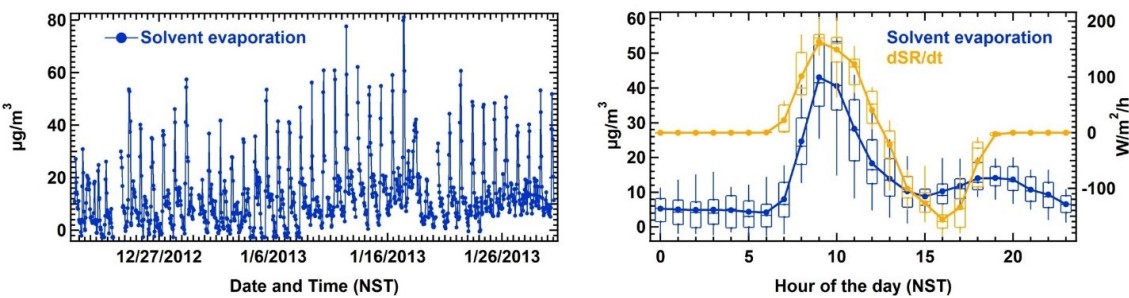

**Figure 9.** Timeseries and diel box and whisker plot for Factor 6 (Solvent evaporation)

diel profiles is shown in Figure S6 of the supplementary information. Since this factor seems to contain contributions of multiple sources and potentially the photooxidation products of their emissions, this factor was termed as the unresolved industrial emissions factor.

Most of the total measured NMVOC mass of Factor 5 was due to oxygenated NMVOCs like methanol ($\sim 14\,\%$), acetic acid ($\sim 11\,\%$), acetaldehyde ($\sim 9\,\%$), acetone ($\sim 9\,\%$) and formic acid ($\sim 9\,\%$) but benzene, propyne and propene also contributed $> 5\,\%$ ($\sim 9\,\%$, $\sim 6\,\%$ and $\sim 6\,\%$, respectively) to the total measured NMVOC mass of this factor. The other 29 NMVOCs together contributed only $\sim 27\,\%$ to this factor and their individual contributions were less than $5\,\%$.

### 3.1.6 Factor 6 - Solvent evaporation

Factor 6 explains approximately $25\text{-}40\,\%$ of the compounds containing the aldehyde functional group. It explained $\sim 39\,\%$ of the total acetaldehyde, $\sim 27\,\%$ of the total formaldehyde and $\sim 23\,\%$ of 2-furaldehyde. Moreover, $\sim 28\,\%$ of the total acetic acid and $\sim 23\,\%$ of the total methylglyoxal were explained by this factor. Acetaldehyde and acetic acid contributed $\sim 40\,\%$ and $\sim 27\,\%$ respectively to the total measured NMVOC mass of Factor 6 while formic acid, formaldehyde, acetone and ethanol together contributed $\sim 15\,\%$ ($\sim 5\,\%$, $\sim 4\,\%$ and $\sim 3\,\%$, respectively) to the total measured NMVOC mass of this factor. The other 31 species contributed only $\sim 18\,\%$. The diel profile (Figure 9) of this factor correlates best with the increase in rates of temperature (dT/dt, $R^2 = 0.41$) and solar

radiation (dSR/dt, $R^2$ = 0.38) during the daytime hours (between 06:00 – 17:00 LT; as can be seen in Table S4 of the supplementary information). Factor 6 showed a sharp peak directly after sunrise between 08:00 – 10:00 LT. This time coincides with the maximum increase in both temperature and solar radiation. Average mass loadings of $\sim 45$ µg m$^{-3}$ were observed during this period. However, the change of the saturation vapor pressure for a temperature change from 5 °C to 20 °C for the dominant compounds (acetaldehyde and acetic acid) is small (less than a factor of 1.3; Betterton and Hoffmann (1988); Johnson et al. (1996)) and, therefore, does not account for the observed magnitude of increase (by a factor of $\sim 5$) from 06:00 - 09:00 LT . Instead, the temperature dependence of the solubility of these compounds in an aqueous solution (factor 5-7) would explain a change of this magnitude. The sharp peaks observed in this factor during morning hours could be explained by the Kathmandu Valley meteorology. After sunrise when air temperatures start to rise, the boundary layer continues to be shallow until direct sunlight reaches the Valley bottom. The accumulation of compounds in a shallow boundary layer contributes to high ambient concentrations. The dilution due to the rising boundary layer and daytime westerly winds in the Valley reduces the concentrations subsequently. Therefore, this factor is attributed as solvent evaporation.

### 3.1.7  Factor 7 - Mixed daytime

Formic acid and acetic acid contributed most to the total measured NMVOC mass of Factor 7 ($\sim$ 25 % and $\sim$ 13 %, respectively) while propyne, methanol and acetone together contributed $\sim 26$ % ($\sim 10$ %, $\sim 8$ % and $\sim 8$ %, respectively). The other 32 species collectively contributed $\sim 36$ % to this factor but their individual contributions were $\leq 5$ %. Like factor 6, this factor, too, has a predominance of oxygenated compounds (that could be due to photooxidation) with a minor contribution from NMVOCs such as acetonitrile and propyne which can be emitted from primary emission sources such as biomass burning and industrial emissions (Hao et al., 1996; Andreae and Merlet , 2001; Akagi et al., 2011). The diel profile of this factor (Figure 10) is similar to that of the ambient temperature and solar radiation with an average mass concentration of $\sim 20$ µg m$^{-3}$ between 12:00 - 14:00 LT.

Approximately 41 % of the total formamide, $\sim 37$ % of the total acetamide and $\sim 40$ % of the total isocyanic acid are explained by this factor. Both formamide and acetamide can be produced by hydroxyl radical initiated photooxidation of primary amines (such as methyl amine) and in turn can photochemically form isocyanic acid through hydroxyl radical mediated oxidation (Roberts et al., 2014; Ge et al., 2011; Sarkar et al., 2016). In addition 34 % of the formic acid and 23 % of the formaldehyde mass was explained by this factor. The timeseries (Figure 10) of this factor showed higher baseline concentrations during second part of the measurement period when primary emissions were higher due to both biomass burning and biomass co-fired brick kiln emissions as described in Sarkar et al. (2016). During this period, influenced strongly by biomass burning sources, specific NMVOCs such as isocyanic acid, formamide and acetamide showed enhancement in their

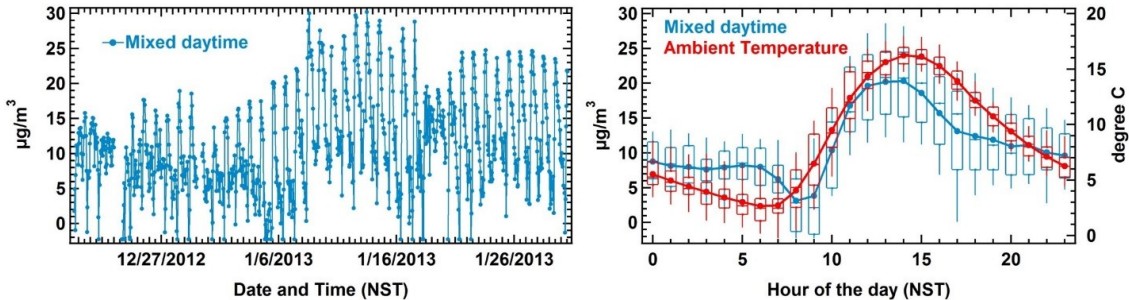

**Figure 10.** Timeseries and diel box and whisker plot for Factor 7 (Mixed daytime)

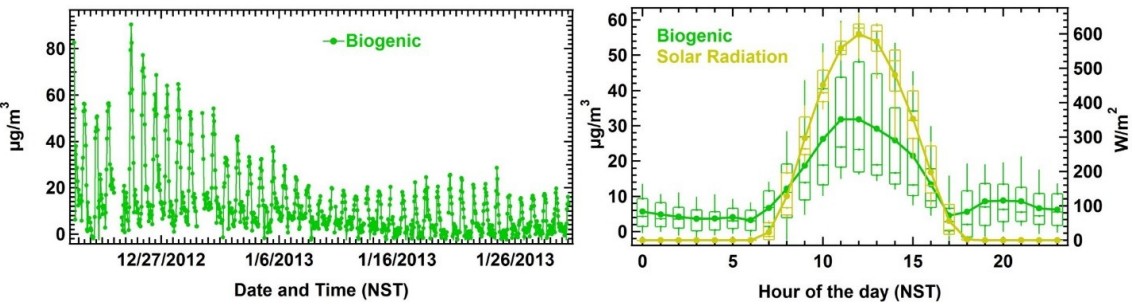

**Figure 11.** Timeseries and diel box and whisker plot for Factor 8 (Biogenic emissions)

background concentrations. This is likely due to the higher emissions of precursor alkyl amines and other N-containing compounds from the incomplete combustion of biomass (Stockwell et al., 2015) which can form formamide and acetamide via photooxidation. Due to the contribution from both
photooxidation and primary emissions, this factor was attributed as the mixed daytime factor.

### 3.1.8  Factor 8 - Biogenic emissions

Factor 8 explains more of the total isoprene mass than any of the other factors ($\sim 33\,\%$) and shows a distinct daytime peak with the highest mass loadings of $\sim 32\,\mu g\,m^{-3}$ observed between 11:00 - 12:00 LT (Figure 11). The diel profile (Figure 11) of this factor correlates best with solar radiation
($R^2 = 0.33$; as can be seen in Table S4 of the supplementary information and Figure S9) during the daytime hours (between 06:00 - 17:00 LT). Average nighttime concentrations of this factor were always less than $10\,\mu g\,m^{-3}$. The timeseries profile showed very high daytime mass loadings up to $\sim 80\,\mu g\,m^{-3}$ for the first part of the campaign (19 December 2012 – 2 January 2013) and lower mass loadings as the campaign progressed. This is also consistent with the observation of deciduous trees
in the Kathmandu Valley shedding their leaves during peak winter (Sarkar et al., 2016). Therefore, the factor was attributed to biogenic emissions.

Most of the total measured NMVOC mass in this factor was associated with oxygenated NMVOCs namely acetaldehyde, acetic acid, acetone and formic acid which contributed $\sim 21\,\%$, $\sim 15\,\%$, $\sim$

11 % and $\sim$ 10 %, respectively to Factor 8. Isoprene contributed $\sim$ 8 % to the total NMVOC mass. The other 32 NMVOCs together contributed $\sim$ 35 %.

To summarize, based on the characteristics observed in the factor profiles, factor timeseries and diel plots, Factor 1 was attributed to traffic (TR), Factor 2 was attributed to residential biofuel use and waste disposal (RB+WD), Factor 3 was attributed to mixed industrial emissions (MI), Factor 4 was attributed to biomass co-fired brick kilns (BK), Factor 5 to unresolved industrial emissions (UI), Factor 6 was attributed to solvent evaporation (SE), Factor 7 was attributed to mixed daytime source (MD) and Factor 8 was attributed to biogenic NMVOC emissions (BG). Table S4 of the supplementary information shows the calculated correlation coefficients between the PMF resolved source factors and the independent meteorological parameters.

It can be seen from Table S4 of the supplementary information that during daytime, the solvent evaporation (SE) factor correlated best with the rate of change in solar radiation and the rate of change in ambient temperature (r = 0.62 and 0.64, respectively). This supports the assignment of the solvent evaporation factor as evaporation depends on temperature. The solvent evaporation factor strongly anti-correlated with RH during the nighttime (R=-0.59)and correlated well with the unresolved industrial (UI) factor (R = 0.55), changes in solar radiation (R=0.62) and $\Delta$T (R=0.64) during daytime. While the correlation of the solvent evaporation factor with the unresolved industrial factor during daytime seems to suggest the two should be combined into one factor profile, several facts suggest against it. Firstly, the two do not correlate at night since the unresolved industrial factor shows a mild positive correlation rather than anti-correlation with RH at night (R=0.29) and no strong correlation with $\Delta$T during the day (R=0.28). Secondly, the raw time series of 1,3-butadiyne and methanol (Supplementary Table S5) correlates extremely strongly (R=0.9), indicating there is a strong and unique common source which causes sharp spikes in these two compounds. The fact that the time series of 1,3-butadiyne correlates poorly with acetaldehyde, acetic acid and formic acid indicates that the solvent evaporation factor (which is not a significant source of 1,3-butadiyne and methanol), has very different emission ratios of 1,3-butadiyne to acetaldehyde, acetic acid and formic acid compared to the unresolved industrial emissions factor to explain the raw data. The fact that the time series of 1,3-butadiyne correlates equally poorly with that of ethanol, the defining compound of the mixed industrial factor, suggest against combining the mixed industrial factor with the unresolved industrial factor. It, therefore, seems, that the unresolved industrial factor is related to primary emissions from a distinct source, while the source profile of the solvent evaporation factor may be strongly confounded by meteorology and chemistry. Confounding factors have been reported to affect PMF solutions previously (Yuan et al., 2012).

The mixed daytime factor (MD) correlated with solar radiation, ambient temperature and wind speed (r = 0.58, 0.74 and 0.57, respectively). The biogenic factor (BG) had the best correlation with solar radiation (r = 0.57) during daytime, consistent with its attribution to biogenic emissions. During daytime, the mixed industrial emissions and biomass co-fired brick kiln emissions had very

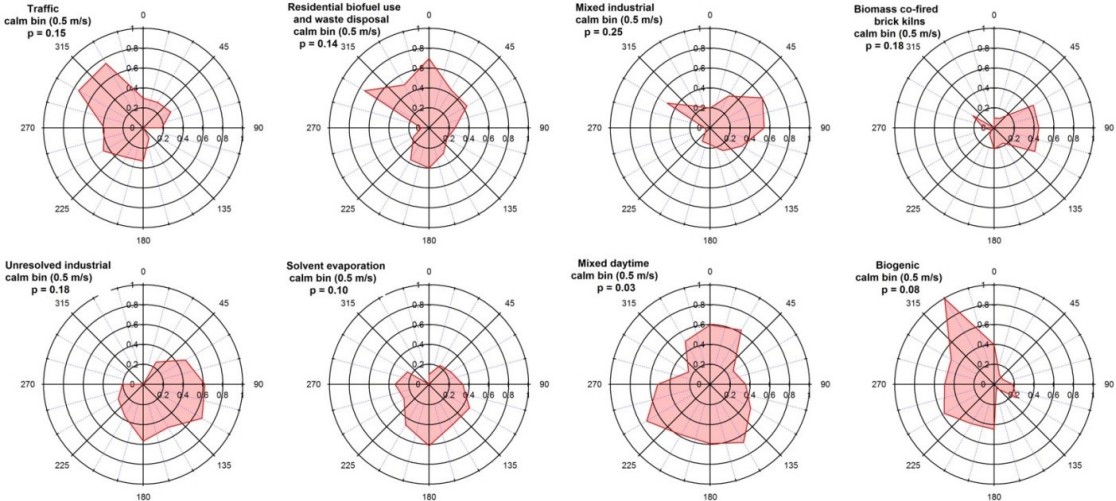

**Figure 12.** Conditional probability functions (CPF) plots for all source factors resolved by PMF showing wind directional dependency of different source categories

low mass concentration due to the boundary layer dilution and ventilation effect of high westerly winds in the Kathmandu Valley (Sarkar et al., 2016). The ambient RH was also lower during the daytime. Therefore, both the mixed industrial emissions and brick kilns emission showed positive correlations with ambient RH (r = 0.65 and 0.74, respectively). During nighttime, no significant correlation was observed between the PMF resolved factors except the correlation of the biogenic factor with the residential biofuel use and waste disposal (RB+WD) factor (r = 0.58) which indicates that the high emissions of oxygenated NMVOCs and isoprene from RB+WD sources could result in a minor mis-attribution of the combustion derived emissions to the biogenic factor.

### 3.2 Conditional probability functions (CPF) to determine source directionality

Figure 12 shows the Conditional Probability Function (CPF) plots that were used to examine the spatial profile of the eight different PMF source factors. For the CPF plots, only data with wind speed $> 0.5\,\mathrm{ms}^{-1}$ were considered. Six factors namely traffic, residential biofuel use and waste disposal, mixed industrial emissions, unresolved industrial emissions, solvent evaporation and biomass co-fired brick kilns could be associated clearly with anthropogenic activities and are, therefore, likely to be impacted by spatially fixed sources, while one factor (mixed daytime) was related to photochemistry. One factor, biogenic emissions, is natural but can also be attributed to spatially fixed sources such as forests.

The CPF plot for the traffic factor showed maximum conditional probability (0.4 - 0.7) from the W-NW direction where the Kathmandu city center and the busiest traffic intersections were located. The conditional probability for the SW and NE wind direction ranged from 0.2-0.4. Two

cities, namely Lalitpur (Patan) and Bhaktapur, respectively, are located upwind of the site in these directions. The lowest conditional probability was observed for the SE wind direction.

The residential biofuel use and waste disposal factor showed a high conditional probability of emissions exceeding the mean for air masses reaching the site from most wind directions (0.5 - 0.7 for NW-N, $\sim 0.4$ for N-NE and S-SW and 0.2 for E-S), indicating that this source is spatially distributed throughout the Kathmandu Valley. Only for the wind sector from SW-NW the conditional probability of this source is low. The reason for this low conditional probability is that every day in the afternoon, winds from the western mountain passes reach the receptor site. The same wind direction is extremely rare after sunset and during the early morning hours, when residential biofuel use and waste disposal mostly occur. Consequently, the conditional probability plot shows low conditional probabilities for this wind sector.

The mixed industrial emissions factor showed the highest conditional probability of air masses with above average mass loadings reaching the receptor site from the NE to SE wind sector (p = 0.4-0.6), where Bhaktapur industrial estate is located within a distance of 3-4 km upwind of the receptor site. Conditional probabilities of 0.2-0.4 were observed for the NW wind direction where several industries are located.

For brick kilns the highest conditional probability was observed for air masses reaching the receptor site from the NE-SE (p $\sim 0.4$), which had several active brick kilns near the Bhaktapur Industrial Estate, which was $\sim 4$ km upwind of the receptor site.

It is interesting to note that the unresolved industrial emissions factor shows a clear directional dependence (p = 0.5-0.7 for the NE-SW wind sector) indicating that this factor, too, can be attributed to spatially fixed sources in Bhaktapur Industrial Estate and Patan Industrial Estate. Polymer production, manufacturing industries for adhesives, paints and/or pharmaceuticals upwind of the site likely contributed towards the measured NMVOC mass of the unresolved industrial factor.

The solvent evaporation factor, too, shows high conditional probabilities for the SE-SW wind direction (Patan Industrial Estate) and low conditional probabilities for the NW-NE wind direction. The conditional probability function shows significant overlap with that of the unresolved industrial emissions factor. It therefore highlights the plausibility that solvent/chemical evaporation or emissions from industrial units are the primary source for this factor although the temperature changes after sunrise drives the partitioning into the gas phase.

Within the bin of calm wind speeds ($< 0.5\,\mathrm{ms^{-1}}$) the maximum conditional probabilities were observed for mixed industrial emissions, unresolved industrial emissions and brick kilns (0.25, 0.18 and 0.18, respectively) which indicates that emissions from these sources tended to accumulate in a shallow boundary layer during stagnant conditions in the Kathmandu Valley. Therefore, using taller chimney stacks, at least for combustion sources, to prevent accumulation of emissions in a shallow boundary layer could potentially improve the air quality of the Valley during foggy nights.

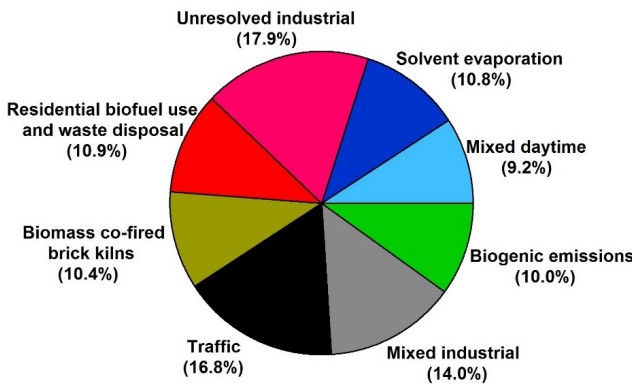

**Figure 13.** Contributions of various sources to the total NMVOC mass loading observed at Bode, a semi-urban site in the Kathmandu Valley

The mixed daytime factor shows no obvious directional dependence for the conditional probability of recording values above the average at the receptor site ($p > 0.3$ for all directions). Slightly higher conditional probabilities ($p \sim 0.6$) are recorded for air masses reaching the receptor site from the N-NE and S-SW wind direction.

The biogenic factor showed high conditional probabilities for air masses reaching the receptor site from the SW to N direction ($p = 0.5$ to $1$) where few forested areas such as Nilbarahi jungle and Gokarna forest were located. Also forested areas in mountain slopes in the SW and NW direction and the midday fetch region being frequently from this sector explains the directional dependency of the biogenic factor.

The CPF analysis of the PMF model output clearly indicates that spatially fixed sources are responsible for a significant fraction of the overall measured NMVOC mass loadings and opens up the possibility to identify and mitigate emissions or at least the build-up of pollutants in a shallow inversion.

### 3.3 Source contribution to the total measured NMVOC mass loading and comparison with emission inventories

Figure 13 shows a pie chart summarizing contributions of individual sources to the total measured NMVOC mass loading. Total measured NMVOC mass loading was calculated by summing up the concentrations of individual measured NMVOCs (in $\mu g\, m^{-3}$). The distribution shows that biogenic sources and the mixed daytime factor contributed only $10\,\%$ and $9.2\,\%$, respectively, to the total measured NMVOC mass loading while all the anthropogenic sources collectively contributed $\sim$ $80\,\%$ to the total measured NMVOC mass loading.

According to two widely used emission inventories, namely REAS v2.1 (Regional Emission inventory in ASia) and EDGAR v4.2 (Emissions Database for Global Atmospheric Research) (Kurokawa et al., 2013; Olivier et al., 1994) and the existing Nepalese inventory obtained from the International Centre for Integrated Mountain Development's (ICIMOD) database, residential biofuel use is considered to be the pre-dominant source of anthropogenic NMVOC emissions in Nepal. When the analysis is spatially restricted to the Kathmandu Valley for those inventories that provide gridded emissions (as shown in Figure 14), differences between EDGAR v4.2 and REAS v2.1 appear.

The EDGAR v4.2 inventory (for the full year 2008) attributes only 10.6% of the total anthropogenic NMVOC emissions in the Kathmandu Valley (85.2-85.5 Longitude and 27.6-27.8 Latitude) to be due to residential biofuel use and an additional 8.9% to solid waste disposal. These numbers are in reasonable agreement with our PMF output, which attributes 13.5% instead of 19.5% of the total measured NMVOC mass to these two sources combined. EDGAR v4.2 inventory provides only spatially resolved data, not seasonally resolved data.

The REAS v2.1 inventory (for the year 2008) estimates that 67.2 % of the total wintertime (December and January) anthropogenic NMVOC emissions in the Kathmandu Valley (85.25–85.5 Longitude and 27.5–27.75 Latitude) originate from residential and commercial biofuel use — a significant overestimation when the numbers are compared to our PMF output and the EDGAR v4.1 inventory. The national Nepali emission inventory, too apportions a large share of the total national annual NMVOCs emissions to residential and commercial biofuel use (83.1 %). It, therefore, appears, that while apportioning the emissions spatially, the REAS v2.1 emission inventory does not fully account for the socio-economic differences between rural and urban areas. The EDGAR v4.2 emission inventory, on the other hand, seems to apportion most of the national consumption of LPG cooking gas to the highly urbanised Kathmandu valley and correspondingly scales down the emission from biofuel use within the Kathmandu valley. In absolute terms the annual NMVOC emissions attributed to domestic fuel usage within the Kathmandu valley by EDGAR v4.2 are a factor of 3.6 lower compared to the annual NMVOC emissions attributed to this sector by REAS v2.1.

The EDGAR inventory considers solvent use (66 %) and mixed industrial emissions to represent the second most important source of NMVOCs. Solvent use and other industrial emissions (8.5 %) combined account for 74.5 %. Collectively they are considered to contribute $\sim$ 10,% to the total anthropogenic NMVOC mass in the EDGAR v4.2 inventory, while our PMF results attribute 52.8 % of the measured NMVOCs to solvent use and industrial emissions combined.It should be noted, that solvent use and other factors related to industrial emissions (mixed industrial and unresolved industrial) must be combined while comparing our PMF output with emission inventories. Both the mixed industrial emission factor and the unresolved industrial emission factor contain a significant NMVOC mass fraction from industrial solvent use, but also combustion related emissions from industrial units. Unfortunately, industrial solvent use and industrial combustion emissions from co-located units cannot be cleanly segregated using the PMF model, which relies on spatio-temporal

patterns while building factor profiles. Overall, our PMF output agrees with the EDGAR v4.2 inventory, that industries are the dominant source of NMVOCs in the Kathmandu valley. According to the REAS v2.1 inventory, solvent use is considered to be the second most dominant contributor (29.8 %) to wintertime NMVOC emissions in the Kathmandu valley. Solvents and other industrial emissions (0.9 %) combined account for 30.7 % of the total wintertime NMVOC emissions in the REAS v2.1

emission inventory. Since, most of the national consumption of solvents and a significant share of Nepal's industrial production is concentrated in the Kathmandu valley, the discrepancies between the REAS v2.1 emission inventory and our results indicate, that the REAS v2.1 emission inventory does not sufficiently account for the special status of the Kathmandu valley while spatially apportioning emissions. The emissions that EDGAR v4.2 attributed to solid waste disposal, industries, the

transport sector, and solvent use within the Kathmandu valley are a factor of 17.4, 14.0, 7.4 and 3.3 times higher compared to what the REAS v2.1 inventory attributes to the same sectors for the same geographical area.

The annual Nepalese inventory (for the year 2000) considers solvent and paint use to be the second largest contributor to the anthropogenic NMVOC emissions in Nepal, while industries are considered

to make an insignificant overall contribution (0.7 %). These numbers cannot be compared to our results in a meaningful manner, as the national emissions in particular for sectors such as domestic fuel usage and agricultural waste burning may be dominated by the rural hinterland, while our PMF results apply to the largest urban agglomeration in Nepal.

Traffic was considered to contribute only between $\sim 1.3$ % (in the REAS v2.1 inventory) to a

maximum of $\sim 2.6$ % (in EDGAR v4.2 inventory) of the total anthropogenic NMVOC emissions in the Kathmandu valley. This stands in stark contrast to the results of our PMF analyses, which indicate traffic contributes ca. 20 %, solvent evaporation and industrial solvent/chemical usage accounts for ca. 36 % (unresolved industrial emissions + solvent evaporation) and other industrial emissions (mixed industrial emissions + brick kilns) account for ca. 30 % of the total measured anthropogenic

NMVOC mass loading in the Kathmandu valley. According to the recent study of the vehicle fleet in Kathmandu valley Shrestha et al. (2013), transport sector NMVOC emissions in the Kathmandu valley for the year 2010 amounted to 7654 $\mathrm{t\,y^{-1}}$, a number that is 10 times higher than the number currently in the EDGAR v4.2 inventory and 72 times higher than the number currently in the REAS v2.1 inventory. If the emission estimate of (Shrestha et al., 2013) was incorporated into EDGAR

v4.2 inventory without any further changes, the percentage share of transport sector emissions to the total NMVOC emissions would increase to 38.7 %, while the contribution of domestic fuel usage and waste disposal would drop to 12.7 %(PMF 13.5 %) and the contribution of industrial emissions and solvent use would drop to 48.6 % (PMF 52.8 %). Our PMF results, however, seem to suggest, that 2012 transport sector emissions have decreased by $\sim$50 % compared to the 2010 emissions pre-

sented in (Shrestha et al., 2013), possibly due to a reduction of the number older vehicles in the fleet.

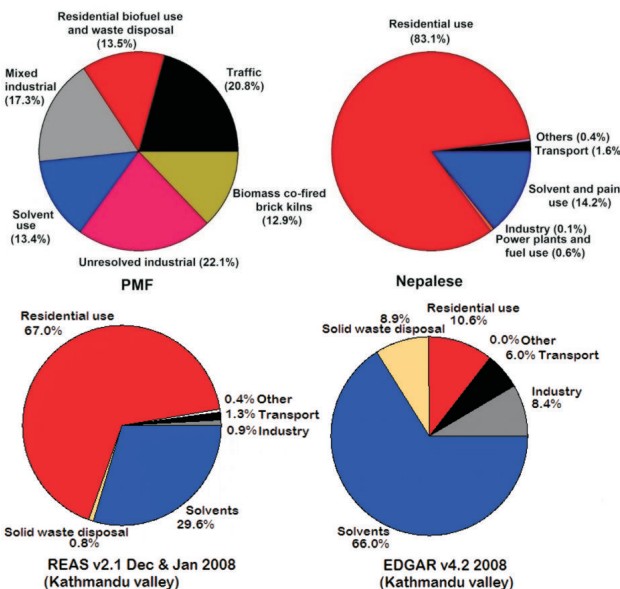

**Figure 14.** Comparison of the PMF derived contribution of anthropogenic sources with NMVOCs source contribution according to the existing Nepalese, REAS and EDGAR emission inventory

Inefficient biomass co-fired brick kilns are a unique industrial source in the Kathmandu Valley, and contributed significantly ($\sim 15\%$) to the total measured anthropogenic NMVOC mass loading. The existing Nepalese inventory considers contributions of brick kilns only to the emission of particulate

matter ($PM_{10}$ and $PM_{2.5}$)), while the two other emission inventories do not include emissions from brick kilns in the Kathmandu Valley at all. If transport sector NMVOC emissions of $\sim 3800\ \mathrm{t\,y^{-1}}$ and an additional $\sim 2400\ \mathrm{t\,y^{-1}}$ NMVOC emissions from brick kilns, were included in the EDGAR v4.2 emission inventory, the EGAR emission inventory and our PMF output would agree perfectly (within $\pm\,0.2\%$) on the relative contribution of all sources, without changing the contribution from

any of the other sources.

Only two sources, domestic fuel usage (on account of the changed heating demand) and agricultural waste burning are expected to have significant seasonality. Jointly, they account for less than $10\%$ of the total NMVOC emissions. Since cooking needs persist throughout the year and the decrease in agricultural waste burning outside harvest season may be partially offset by leaf-litter

burning (a source currently not in the model), it is likely that the failure to account for seasonal effects imparts an uncertainty of less than $1\%$ on the overall result of our analysis.

The REAS v2.1 emission inventory for the Kathmandu valley, on the other hand, seems to require large corrections. While our analysis of the REAS inventory was restricted to December and January, annual averages of individual sources differ by less than $\pm\,10\%$ from the winter values. Therefore,

the difference in the time window selected for the analysis cannot explain the observed discrepancies to the EDGAR emission inventory.

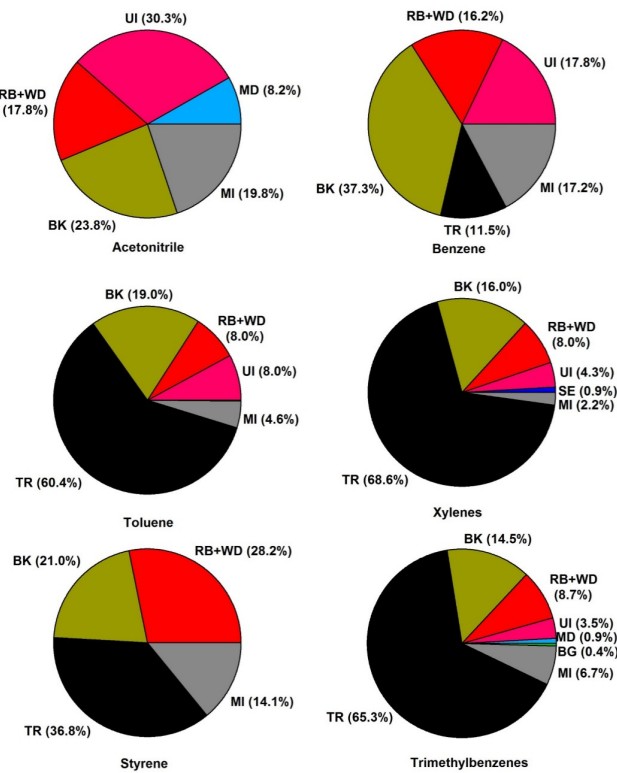

**Figure 15.** Contribution of PMF derived source factors to acetonitrile and aromatic NMVOCs. Source names are abreviated as follows: MD=mixed daytime, MI= mixed industrial, UI = unresolved industrial, BK = brick kiln, TR = traffic, RB+WD = residential burning and waste disposal, SE = solvent evaporation, BG = biogenic

### 3.4 Source contribution to individual NMVOCs

Figure 15 represents the pie charts showing contribution of the eight source factors to individual NMVOCs such as acetonitrile, benzene, styrene, toluene, sum of C8-aromatics (xylenes and ethyl-
benzene) and sum of C9-aromatics (trimethylbenzenes and propylbenzene). Maximum contribution to the acetonitrile mass concentration was observed from the unresolved industrial emission sources ($\sim 30\,\%$) followed by the biomass co-fired brick kilns emission ($\sim 24\,\%$) and mixed industrial emission ($\sim 20\,\%$) factors. Residential biofuel use and waste disposal features only fourth ($\sim 18\,\%$). The same sources also contribute most to benzene emissions, indicating that fuel usage, rather than its
application as solvent/chemical reagents in industrial processes is responsible for most of the industrial acetonitrile emissions. It also indicates that industrial rather than residential biofuel usage contributes more towards outdoor NMVOC air pollution. Most of the benzene (which is a human carcinogen) can be attributed to biomass co-fired brick kilns ($\sim 37\,\%$), mixed industrial ($\sim 17\,\%$) and unresolved industrial ($\sim 18\,\%$) sources. Residential biofuel use again featured only fourth as
far as the contribution towards mixing ratios of this compound in the outdoor environment is con-

**Table 6.** Emission ratios of NMVOCs/benzene for acetonitrile and aromatic hydrocarbons derived from the PMF model for different sources and comparison with the ratios for different source categories reported in previous studies.

| ERs/Benzene | RB+WD | BK | MI | UI | Garbage burning grab samples | Waste burning[1] | Wood burning[2] | Charcoal burning[2] |
|---|---|---|---|---|---|---|---|---|
| Acetonitrile | 0.23 | 0.14 | 0.25 | 0.36 | 0.77 | 0.06 | - | - |
| Toluene | 0.34 | 0.35 | 0.18 | 0.30 | 0.34 | 0.41 | 0.05 | 0.50 |
| C8-aromatics | 0.18 | 0.06 | 0.08 | 0.00 | 0.25 | 0.10 | - | 0.46 |
| C9-aromatics | 0.25 | 0.22 | 0.06 | 0.12 | 0.08 | 0.03 | - | - |
| Styrene | 0.12 | 0.09 | 0.09 | 0.04 | 0.16 | 0.86 | - | - |
| Naphthalene | 0.11 | 0.15 | 0.20 | 0.05 | 0.09 | 0.10 | - | - |

1. Stockwell et al. (2015); 2. Tsai et al. (2003); RB+WD = Residential biofuel use and waste disposal; BK = Biomass co-fired brick kilns; MI = Mixed industrial emissions; UI = Unresolved industrial emissions

cerned. Table 6 shows a comparison of NMVOCs/benzene emission ratios for four PMF derived sources (residential biofuel use and waste disposal, biomass co-fired brick kilns, mixed industrial and unresolved industrial sources) to the emission ratios obtained from the grab samples collected for garbage burning in the Kathmandu Valley and the previously reported emission ratios for waste burning, wood burning and charcoal burning sources.

Residential biofuel use and waste disposal contributed $\sim 28\,\%$ of the total styrene which were emitted significantly from waste burning. However, traffic was found to be equally important as a styrene source ($\sim 37\,\%$) in the Kathmandu Valley. Recently, styrene has been detected from traffic and was found to have high emission ratios with respect to benzene after cold startup of engines and in LPG fuel (Alves et al., 2015). Biomass co-fired brick kilns and mixed industrial emissions also contribute significantly ($\sim 21\,\%$ and $\sim 14\,\%$, respectively) towards styrene mass loadings. Traffic was found to be the most important source of higher aromatics including toluene, C8-aromatics, and C9-aromatics ($> 60\,\%$). Biomass co-fired brick kilns were the second largest contributors towards their mass loadings, while residential biofuel usage and waste disposal ranked third.

Figure 16 shows the pie charts summarizing contributions of PMF derived sources to two newly quantified compounds in the Kathmandu Valley, namely formamide and acetamide along with iso-cyanic acid and formic acid. All these compounds showed maximum contribution from the mixed daytime factor ($\sim 34\,\%$ to $\sim 41\,\%$) due to the photo-oxidation source. As discussed previously in Sarkar et al. (2016) and in section 3.1.7, both formamide and acetamide are formed primarily as a result of photooxidation of amine compounds and N-containing compounds. These can be emitted from the various inefficient combustion processes in the Kathmandu Valley. Photooxidation of these amides further forms isocyanic acid (reaction schematic is shown in Figure S8 of the supplementary information). Apart from the mixed daytime source, unresolved industrial emissions factor also

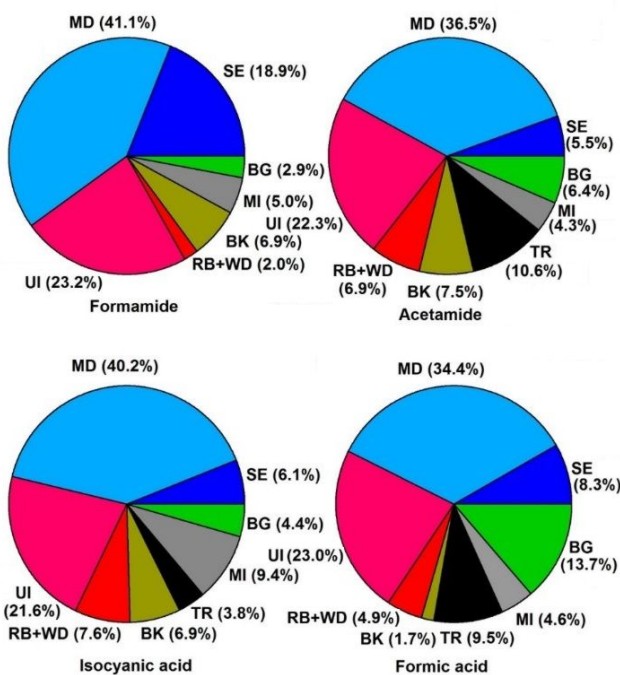

**Figure 16.** Contribution of PMF derived sources to formamide, acetamide, isocyanic acid and formic acid. Source names are abreviated as follows: MD=mixed daytime, MI= mixed industrial, UI = unresolved industrial, BK = brick kiln, TR = traffic, RB+WD = residential burning and waste disposal, SE = solvent evaporation, BG = biogenic

contributed significantly to all these compounds ($\sim 22\,\%$ to $\sim 23\,\%$) as they are used as reactants
(e.g. formic acid is used as reactant to produce formamide in industries) or produced during different industrial processes (such as formamide is produced in pharmaceuticals and plastic industries ). Solvent evaporation factor contributed $\sim 19\,\%$ to formamide while biogenic factor contributed $\sim 14\,\%$ to formic acid. Contributions from all the other sources to these NMVOCs were $< 10\,\%$.

Figure 17 represents the pie charts showing contribution of the eight sources derived from PMF
to 1,3-butadiyne and oxygenated compounds namely methanol, acetone, acetaldehyde, ethanol and acetic acid. It can be seen from Figure 17 that emissions of all these compounds in the Kathmandu Valley were dominated by different industrial activities. The total unresolved industrial emissions factor dominated the contribution to 1,3-butadiyne ($\sim 48\,\%$), methanol ($\sim 35\,\%$) and acetone ($\sim 22\,\%$). Residential biofuel use and waste disposal also contributed significantly to 1,3-butadiyne
($\sim 21\,\%$) and methanol ($\sim 16\,\%$). Traffic was found to have significant contribution to acetone ($\sim 21\,\%$). It is known that acetaldehyde, ethanol and acetic acid are used as solvents in different industries and it was found that industrial sources obtained from PMF (mixed industrial + unresolved industrial + solvent evaporation) together contributed $\sim 72\,\%$ of the total acetaldehyde, $100\,\%$ of the

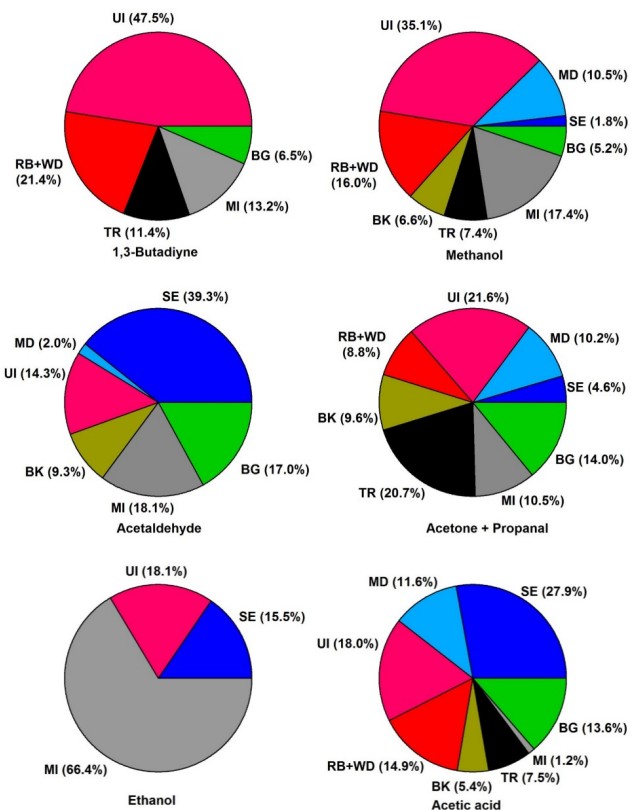

**Figure 17.** Contribution of PMF derived sources to 1,3-butadiyne and oxygenated NMVOCs such as methanol, acetone, acetaldehyde, ethanol and acetic acid. Source names are abreviated as follows: MD=mixed daytime, MI= mixed industrial, UI = unresolved industrial, BK = brick kiln, TR = traffic, RB+WD = residential burning and waste disposal, SE = solvent evaporation, BG = biogenic

total ethanol and $\sim 47\%$ of the total acetic acid. Biogenic sources also had significant contribution
to acetaldehyde and acetic acid ($\sim 17\%$ and $\sim 14\%$, respectively) whereas residential biofuel use
and waste disposal contributed to $\sim 15\%$ of the total acetic acid.

Figure 18 represents a timeseries of daily mean relative contribution of the PMF derived sources
during SusKat-ABC campaign. As discussed in Sarkar et al. (2016), the whole campaign can be
divided into three different periods based on the measurements – first period (from the start of the
campaign until 3 January 2013) was associated with high daytime isoprene emissions due to strong
biogenic emissions, the second period (4 – 18 January 2013) was marked by enhancements in ace-
tonitrile and benzene concentrations due to the kick start of the biomass co-fired brick kilns in the
Kathmandu Valley and in the third period (19 January until the end of the campaign), more oxy-
genated NMVOCs were observed which was believed to be due to the stable operation of the brick
kilns and more contribution from the industrial sources. PMF derived results also supports these
observation as can be seen in Figure 18. It can be seen that from the start of the campaign until

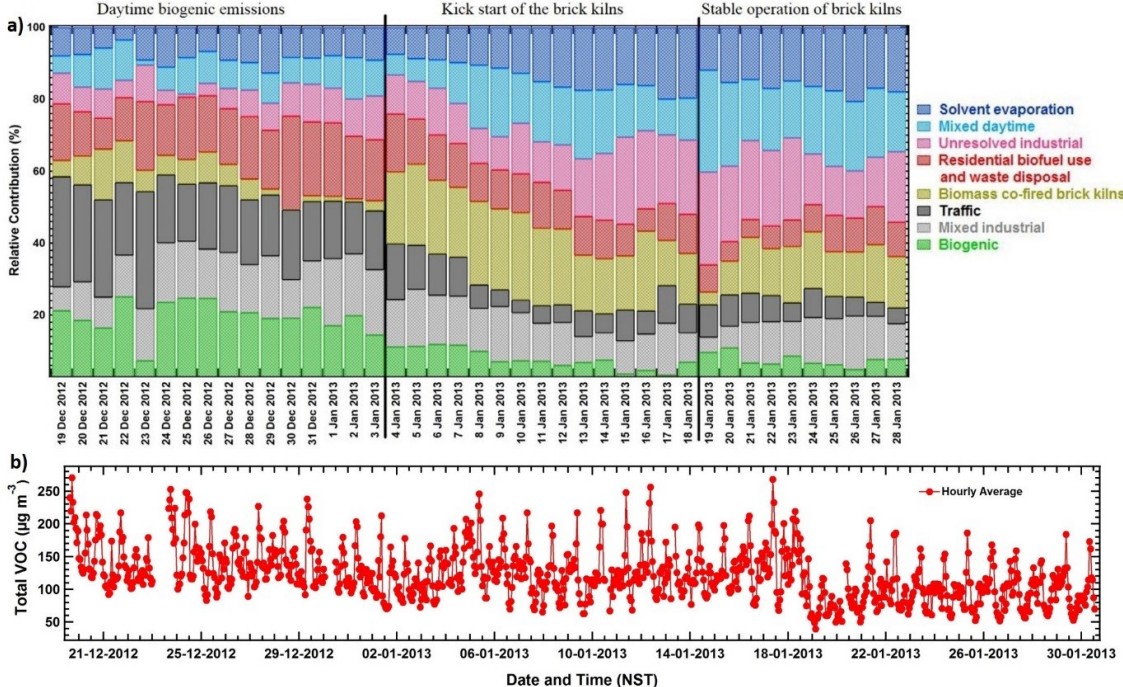

**Figure 18.** Daily mean relative contribution of PMF derived eight sources during SusKat-ABC campaign

3 January 2013 contribution of PMF derived biogenic sources were $> 20\%$ for most of the time while contribution from the brick kilns emission factor was negligible ($\leq 5\%$). From 4 January until 18 January 2013, the contribution of brick kilns increased significantly ($\sim 20\%$ to $\sim 40\%$) as almost all brick kilns in the Kathmandu Valley became operational. After 18 January until the end of the campaign, the contribution of brick kilns become lower due to its stable operation.

During the first period, contribution of traffic was found to be higher ($\sim 20\%$ to $\sim 30\%$) compared to the rest of the campaign. The higher contribution of the mixed daytime source during the second and third part of the campaign was due to the early morning and daytime photooxidation of the precursor compounds which were emitted as a result of biomass co-fired brick kilns and other biomass burning emissions during these periods. The mixed industrial emissions factor contributed almost equally throughout the campaign (contributing $\sim 10\%$ to $\sim 15\%$) but the solvent evaporation and the unresolved industrial emissions factor contributed more during the second and third part of the campaign (increase of $\sim 10\%$).

### 3.5 Source contribution to daytime ozone production potential and SOA formation

Figure 19a represents the source contribution to daytime $O_3$ production potential while Figure 19b represents the contribution of different classes of compounds measured in the Kathmandu Valley to the daytime $O_3$ production potential as discussed in Sarkar et al. (2016). The daytime $O_3$ production

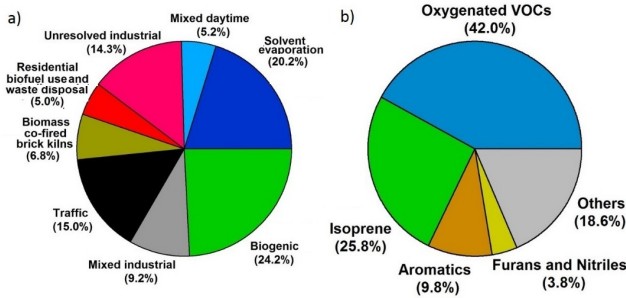

**Figure 19.** Daytime $O_3$ production potential obtained a) from the source contribution using PMF and b) from the measurements performed in the Kathmandu Valley

potential for individual sources was calculated by summing up the $O_3$ production potential for the in-
dividual compounds which was calculated according to the method described by Sinha et al. (2012).
The distribution of the daytime $O_3$ production potential obtained from the measurements (Figure
19b) shows that $\sim 70\%$ of the total daytime $O_3$ production potential was due to the contribution
from isoprene and oxygenated NMVOCs which could presumptuously indicated dominance of bio-
genic emissions and photochemistry in the Kathmandu Valley even in the winter. But the distribution
of different sources obtained from PMF to daytime $O_3$ production potential shows that the biogenic
factor together with the photochemistry factor (mixed daytime) contributed only $\sim 30\%$ of the total
$O_3$ production potential. The remaining $\sim 70\%$ was contributed by anthropogenic sources. While
solvent evaporation contributed most ($\sim 20\%$) to the total daytime $O_3$ production potential, traffic
and unresolved industrial emission stood second and third, respectively, in terms of anthropogenic
ozone precursor emissions. Residential biofuel use and waste disposal, and biomass co-fired brick
kilns while potentially important from a human health perspective, contributed only a minor fraction
of the total anthropogenically emitted ozone precursors.

The consequence of including only a subset of NMVOCs is an underestimation of the OH reactiv-
ity and hence ozone production potential, which scales directly with the OH reactivity. For the city of
Lahore, Barletta et al. (2016), reported the maximum contribution of methane and 63 non methane
hydrocarbon to the measured OH reactivity as 14%. Lahore is a much larger, and by all indications
more polluted city, than Kathmandu. Despite high concentration abundances in urban atmospheric
environments, the rate constants of these species are typically 100 times lower than compounds like
isoprene, and hence their contribution to the total OH reactivity is much lower. For example, even
3 ppm methane (observed only in plumes) would contribute only $\sim 0.5$ s$^{-1}$ to the total OH reac-
tivity and hence make an insignificant contribution to the ozone production potential. Hence, our
analyses of the ozone production potential may underestimate the total ozone production potential
by 15–25%, if we can extrapolate the observations from another South Asian city like Lahore.

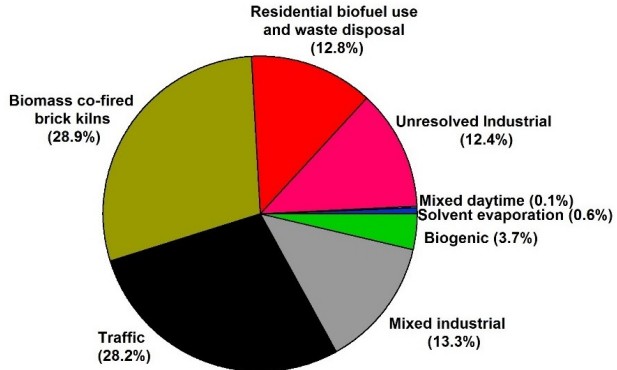

**Figure 20.** Contribution of PMF derived eight sources to the SOA formation in the Kathmandu Valley

Secondary organic aerosol (SOA) production was calculated using the concentrations and the
known SOA yields for benzene, toluene, styrene, xylene, trimethylbenzenes, naphthalene and iso-
prene (Ng et al., 2007; Chan et al., 2009; Yuan et al., 2013; Kroll et al., 2006). As the biomass
co-fired brick kilns and the traffic factor contains most of the reactive aromatic compounds, they ap-
peared to be the dominant contributors to SOA production (as shown in Figure 20) in the Kathmandu
Valley.

## 4 Conclusions

The PMF model results reveal several new results regarding the source apportionment of NMVOCs
in the Kathmandu Valley. Speciation of NMVOCs in the emission inventory for Nepal only includes
compound classes (e.g. alkanes, alkenes etc.) and not specific compounds. This imposes certain lim-
itations while comparing emission inventories with the compounds measured in our study. However,
the existing emission inventories (e.g. REAS v2.1, EDGAR v4.2; Kurokawa et al. (2013); Olivier
et al. (1994) and Nepalese inventory (ICIMOD)) are highly uncertain as there has been no validation
using in-situ measurements of these mostly bottom up inventories which rely on fuel and source
emission factors measured in other technologically different regions of the world (primarily the US
and Europe). By using the specific NMVOC emission tracer data measured in the Kathmandu Valley
and constraining the PMF with measured source profiles of complex sources (e.g. biomass co-fired
brick kilns, residential solid biofuel use and waste disposal), it is shown that the contribution from
sources such as residential solid biofuel use and waste disposal is overestimated in the REAS v2.1
emission inventory. At the same time, the emissions from industrial sources are underestimated.
Both REAS v2.1 and EDGAR v4.2 underestimate the contribution of traffic and do not include brick
kiln emissions. The presence of elevated concentrations of several health relevant NMVOCs (e.g.
benzene) could be attributed to the biomass co-fired brick kiln sources. Eight different NMVOC

sources were identified by the PMF model using the new "constrained model operation" mode. Unresolved industrial emissions (17.8 %), traffic (16.8 %) and mixed industrial emissions (14.0 %) contributed most to the total measured NMVOC mass loading, while biogenic emissions (24.2 %), solvent evaporation (20.2 %), traffic (15.0 %) and unresolved industrial emissions (14.3 %) were the most important contributors to the ozone formation potential. Biomass co-fired brick kilns and traffic contributed approximately equally to the secondary organic aerosol (SOA) production (28.9 % and 28.2 %, respectively), while the most important contributors to the mass loadings of carcinogenic benzene were brick kilns (37.3 %), unresolved industrial (17.8 % and mixed industrial (17.2 %) sources. Photo-oxidation (mixed daytime factor) contributed majorly to two newly identified ambient compounds namely, formamide (41.1 %) and acetamide (36.5 %) along with their photooxidation product isocyanic acid (40.2 %).

This study has provided quantitative information regarding the contributions of the major NMVOC sources in the Kathmandu Valley. This will enable focused mitigation efforts by policy makers and practitioners to improve the air quality of the Kathmandu Valley by reducing emissions of both toxic NMVOCs and formation of secondary pollutants. The results will also enable significant improvements in existing NMVOC emission inventories so that chemical-transport models can be parameterized more accurately over the South Asian region and the air quality-climate predictions by models can become more reliable.

**Authors' contributions**

Sections of this study were submitted in part fulfilment of the PhD work of C.S. carried out under the supervision of V.S. at IISER Mohali. The VOC dataset QA/QC and analyses were performed by C.S. and V.S. whereas B.S. designed and set up the PMF model and ensured QA/QC of PMF output which was performed by C.S. A.P. helped with interpretation of PMF results and suggested grab sampling experiments at an early stage. C.S., V.S. and B.S. wrote the paper and all co-authors discussed the results and commented on the paper.

*Acknowledgements.* Chinmoy Sarkar and Vinayak Sinha acknowledge the support extended by the Founding Director of IISER Mohali, Professor N. Sathyamurthy to enable participation of the IISER Mohali team in the SusKat-ABC campaign. Chinmoy Sarkar acknowledges the Ministry of Human Resources and Development (MHRD), India and IASS Potsdam, Germany for funding with a service contract. IASS Potsdam funded the deployment of the PTR-TOF-MS by the IISER Mohali team in Kathmandu and local logistical support was provided by Khadak S. Mahata, Dipesh Rupakheti, Bhogendra Kathayat at the Bode site.

This study was partially supported by core funds of ICIMOD contributed by the governments of Afghanistan, Australia, Austria, Bangladesh, Bhutan, China, India, Myanmar, Nepal, Norway, Pakistan, Switzerland, and the United Kingdom.

All the data reported in this article can be obtained from the corresponding author by sending an email to vsinha@iisermohali.ac.in.

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
