# Peer review of "Manuscript prepared for J. Name with version 2015/04/24 7.83 Copernicus papers of the LATEX class copernicus.cls. Date: 24 May 2017"

_Atmospheric Chemistry and Physics, 2016_

## Referee Comment (RC1) · Anonymous Referee #1 · 27 Mar 2017

Summary: The authors conducted VOC measurements using PTR-ToF in Kathmandu, Nepal. PMF was used to separate various source contributions to ambient VOC as a function of time. The authors then used a PMF "nudging" tool and some a priori knowledge of source profiles to move the PMF solution into a more physically realistic space. The various PMF factors are identified by comparing their VOC composition with known sources, and their diurnal behavior. The contribution of each PMF factor to total VOC mass is compared to comparable sources in several emissions inventories. The PMF-derived source contributions are quite different from the emissions inventories, which are also quite different from one another. Contributions of the different sources to VOC

mass, O3 formation potential, and SOA potential are discussed.

This is a thorough, detailed, well-written manuscript that provides valuable new information about an important, but underreported, region of the world. I highly recommend publication in ACP, if the following revisions are considered:

General comments: A general concern is that some important information about the data and PMF implementation are missing. I understand that many of the details of data collection and quality assurance have been published elsewhere (Sarkar et al., 2016). Nonetheless, this paper should be able to stand on its own. Several basic pieces of information should be included. For example: - a small map showing the geographical context of the measurement site - a brief description of the instrument & its measurement capability. - a list of the 37 ions used as PMF input, and the reasoning behind their selection. On a similar note, in many locations values for PMF input or assessment are given, but there is no explanation for why these particular values are chosen. (For example: bootstrapping factor assignment with R>0.6; Line 219- Why this particular length of time?). Could you provide a reason for selecting these values, or a literature citation?

Specific comments: Section 2.1 This section needs some minor reorganization. A brief description of the PTR-ToF measurement should come first, then the description of the grab-sampling, then the PMF implementation. I suggest this because the PTR-ToF measurements, and the grab-samples, are referred to several times in the discussion of PMF; however, they had not yet been introduced.

Line 127 I found this sentence very hard to parse. Perhaps you can break this down to provide a clearer explanation of the information provided by bootstrapping.

Lines 127-139 This section could benefit from literature citations describing the use of bootstrapping. I'd also like to see a citation supporting your assertion that a fraction <20% of unmapped factors indicates low random error.

Lines 155-160 This explanation of rotational ambiguity is a little convoluted. Can you rephrase to make this easier to follow? See Ulbrich et al. (2009) as an example.

Line 192 What does it mean to be classified as a "weak" vs. a "strong" species?

Line 193 The conversion from mixing ratio to mass concentration introduces additional variability due to meteorology. Additionally, higher molecular weight species now have more "pull" in the PMF solution, because their signals are higher. Do you see evidence for meteorological influence in the PMF solution? Any evidence of a bias towards explaining variability of heavy species, at the expense of light VOCs? Why did you choose to run PMF on the mass concentration data rather than mixing ratio data?

Lines 200-220 On my first read-through, the presentation of the eight factors was very sudden and it wasn't at all clear to me how their identifications were derived, or that more information would be provided later in the text.

It would be very helpful to see a plot of Q/Qexpected as a function of number of factors, and an additional plot showing what happens to each of your identified factors as the number of factors is changed (perhaps in the supplemental information).

Eight factors is a quite large number compared to PMF solutions reported in many other studies. Without strong supporting evidence for each factor, I find it hard to believe that PMF can robustly separate this many distinct sources. It is especially hard to believe when each factor, on average, accounts for 12.5% of signal – but you have stated that the overall measurement uncertainty is 20% (line 190). Any additional information that you can provide to show that an 8 factor solution is physically plausible would be very helpful in convincing the reader of your solution. This could include additional PMF diagnostics (as suggested above) or other information. For example, it seems that prior to running PMF, you had some idea of what important emission sources to expect, perhaps from your previous paper or from an emissions inventory (I see four listed in the introduction). Can you make a stronger connection between the a priori knowledge and the PMF solution?

Line 204 You discarded solutions with 7 or fewer factors because there appeared to be "mixing" of sources. But, this could also be due to rotational ambiguity. Did you attempt to unmix solutions with ≤7 factors by exploring FPEAK, for example?

Line 226 Can you provide a scatterplot showing $R^2$ between each factor time series, vs $R^2$ between each factor profile? It isn't clear to me if you are discussing correlation between time series or profile here, and it would help to see the "whole picture".

Lines 237-240 Not sure I understand this. Is "mixed daytime" a photochemistry tracer? Why wouldn't solvent evaporation contain acetonitrile or aromatics? Both acetonitrile and aromatics are commonly found in solvents. Why use a ratio to acetic acid as the nudging control?

Figure 2 To which axis do the gray bars and red lines-and-markers belong, respectively?

Line 370 You are interpreting m/z 69 $C_5H_8H^+$ as isoprene, correct? I suggest that this ion mass is actually a cycloalkane or alkane fragment, which seems far more plausible for vehicle exhaust. See Gueneron et al. (2015).

Line 477 Why would solvent evaporation correlate with the rate of change of temperature/sunlight, and not directly with temperature?

Lines 545-548 Can you state (or reiterate, possibly I missed this above) why it cannot be that the solvent evaporation and unresolved industrial factors are "split" from a single source by the PMF? This section also seems fairly complex and highly speculative. Can you cite an example where such a situation has been shown to occur?

Lines 600-610 This also seems to point to a "splitting" of a single source into the unresolved industrial and solvent evaporation factors.

Figure 17 Can you also include the time series of the total VOC mass loading?

Line 818 Many of the oxygenated VOCs are direct emissions from solvents, industry

etc (Figure 16). So I do not think it is correct to say that photochemically produced VOCs are a dominant source of O3 potential, especially when Figure 18a shows that the mixed daytime source contributed only about 5%.

Lines 847-848 Can you clarify how this is related to a result of your work.

Conclusion The conclusion is heavily weighted towards comparison with the emission inventories. While this is an important result, it is not the only finding discussed in the paper. The conclusion could be improved by an assessment of the major findings related to the source contributions to different categories of VOCs, specific VOCs, and O3 and SOA formation potential.

Minor comments (typographical corrections):

Lines 140-145 Put all verbs in present tense for consistency ("will provide" → "provides")

Line 203 "Fewer" than 7 factors

Line 235 "constraints"

Line 289 "FCBTBK": what does this acronym stand for?

Figures 15, 16 Can you include the explanation for the different source acronyms (e.g. "MD, SE, UI") in the caption.

References

Gueneron, M., Erickson, M. H., VanderSchelden, G. S., and Jobson, B. T.: PTR-MS fragmentation patterns of gasoline hydrocarbons, Int. J. Mass Spectrom., 379, 97-109, http://dx.doi.org/10.1016/j.ijms.2015.01.001, 2015.

Sarkar, C., Sinha, V., Kumar, V., Rupakheti, M., Panday, A., Mahata, K. S., Rupakheti, D., Kathayat, B., and Lawrence, M. G.: Overview of VOC emissions and chemistry from PTR-TOF-MS measurements during the SusKat-ABC campaign: high acetaldehyde,

isoprene and isocyanic acid in wintertime air of the Kathmandu Valley, Atmos. Chem. Phys., 16, 3979-4003, 10.5194/acp-16-3979-2016, 2016.

Ulbrich, I. M., Canagaratna, M. R., Zhang, Q., Worsnop, D. R., and Jimenez, J. L.: Interpretation of organic components from Positive Matrix Factorization of aerosol mass spectrometric data, Atmos. Chem. Phys., 9, 2891-2918, 10.5194/acp-9-2891-2009, 2009.

---

## Referee Comment (RC2) · Anonymous Referee #2 · 12 Apr 2017

General comment:

The manuscript shows results of a source apportionment study of NMVOCs measured by PTR-TOF-MS in the Kathmandu Valley in Nepal during winter 2013. Positive matrix factorization analysis was conducted to identify possible emission sources for 37 m/z measured by PTR-MS. The sources were identified from the chemical fingerprint of each PMF factor and their diurnal profiles. Conditional probability functions plots were used to determine the directions of the sources and attribute the chemical emissions to specific spatial areas in the region and specific activities. The sources found by

the authors through PMF were compared with the results of current emission inventories used for Nepal, which, in contrast to the authors results, rely on sources emission factors measured in other regions of the world and are not supported by in-situ collected measurements. Sources and species contributions differ among the authors results and the current inventories as well as between different inventories. Finally, the atmospheric impact as daytime ozone production and SOA formation based on the measured compounds and PMF sources contributions is briefly discussed. I found the manuscript very interesting, of high quality and of high impact as it presents several new findings which can help mitigating the emissions in the region under study. The presented topic also follows in the scope of ACP. The article is overall well written, and results are presented clearly with figures and tables. I highly recommend the manuscript publication, once these specific comments have been addressed:

Specific comments:

-It is a bit confusing how the methods section is presented. There should be a small section introducing the measurement site, the PTR-MS data used, and the grab samples, before any PMF discussion. This would be helpful to follow better section 2.2 and support the nudging tool. Could you list the m/z from PTR-MS you used for implementing the PMF and why? Could you also provide some references of the nudging procedure?

-PTR-TOF-MS usually provides an unambiguous identification of chemical species, however, it would be interesting to know briefly, the operational settings of PTR-MS, which m/z were selected for running the PMF and how the m/z were attributed to the chemical compounds. Were the grab samples measured with the same PTR-MS? Could you provide some information about these data: m/z selected and how the compounds were identified. Line 331, could you provide the standard deviation for the signal stability?

-It would be interesting to provide some details about the calculations for ozone and

SOA formation, you could do this with a short section in the methods after section 2.4.

-Figure 2. The contribution of propyne compared to isoprene for the biogenic factor is quite high, can you comment it?

-Figure 3. How do you explain the higher background and general higher peaks during the first part of the campaign? L. 370: Could you provide more information about isoprene emission from traffic? Could you have any interference on the PTR-MS m/z attributed to isoprene?

-Could be there a connection between oxidation products of traffic emission and the unresolved industrial emissions, as for mixed daytime emissions as oxidation products from biogenic emissions?

-Why biogenic emissions are higher during the first part of the campaign? Were temperature and solar radiation also higher for this part of the campaign?

-Section 3.2 would be easier to follow with a map of the measurement site and mentioned cities, industrial areas and forests.

-Section 3.3, The differences between the current inventories used in Nepal are briefly mentioned in the text, however, it would be interesting to write a few lines at the beginning of this section to introduce the inventories and on which data and assumptions they are based on. Is EDGAR v4.2 also considered for the winter season?

- Lines 786-808, Did you compare your NMVOCs data with the wind directions? How was the wind direction affecting the sources emissions captured at the measurements site?

-Line 815 and 840, please give the equations used for O3 and SOA formation with respective references. It is not easy to understand figure 18 without any specification on the compounds used for each pie chart. Were the measured data used for pie b) the same data sets used to run the PMF?

-Line 828-832, much information is provided, please rephrase the period. Conclusions: Please include a short summary of the main findings here.

Technical comments: -Some acronyms are not explained, or only explained once in the whole manuscript. Could you also provide the extended form of all acronyms used for tables and figures in their captions?

-L. 698, ca. 30%.

---

## Author Comment (AC1) · 23 May 2017

**Source apportionment of NMVOCs in the Kathmandu Valley during the SusKat-ABC international field campaign using positive matrix factorization**

by Chinmoy Sarkar et al., 2017 (ACPD)

**REFEREE 1:**

**Summary: The authors conducted VOC measurements using PTR-TOF in Kathmandu, Nepal. PMF was used to separate various source contributions to ambient VOC as a function of time. The authors then used a PMF "nudging" tool and some a priori knowledge of source profiles to move the PMF solution into a more physically realistic space. The various PMF factors are identified by comparing their VOC composition with known sources, and their diurnal behavior. The contribution of each PMF factor to total VOC mass is compared to comparable sources in several emissions inventories. The PMF derived source contributions are quite different from the emissions inventories, which are also quite different from one another. Contributions of the different sources to VOC mass, $O_3$ formation potential, and SOA potential are discussed. This is a thorough, detailed, well-written manuscript that provides valuable new information about an important, but underreported, region of the world. I highly recommend publication in ACP, if the following revisions are considered:**

We thank the referee for appreciating and highlighting the importance of the work and for highly recommending the manuscript for the publication in ACP subject to revisions. We have found several of Referee 1's comments and suggestions very helpful and these are now reflected in the revised submission (changes are specified in replies and manuscript version with "tracked changes" given at the end of the responses here).

**General comments: A general concern is that some important information about the data and PMF implementation are missing. I understand that many of the details of data collection and quality assurance have been published elsewhere (Sarkar et al., 2016). Nonetheless, this paper should be able to stand on its own. Several basic pieces of information should be included. For example: - a small map showing the geographical context of the measurement site - a brief description of the instrument & its measurement capability. - a list of the 37 ions used as PMF input, and the reasoning behind their selection. On a similar note, in many locations values for PMF input or assessment are given, but there is no explanation for why these particular values are chosen. (For example: bootstrapping factor assignment with R>0.6; Line 219- Why this particular length of time?). Could you provide a reason for selecting these values, or a literature citation?**

We appreciate the reviewer's comment and suggestions. As per the referee's suggestions, we have modified the Materials and Methods (Section 2), by adding two new subsections and moving Section 2.3 pertaining to collection of grab samples, to precede the existing Section 2.1: Positive Matrix Factorization. The two new sections contain a description of the measurement site (as revised Section 2.1) and a description of the PTR-TOF-MS instrument as well as the list of 37 ions measured (as revised Section 2.2) in the revised manuscript. The existing Section 2.1 is now re-numbered as Section 2.4 and other Sections remain as before:

PMF model (as section 2.5) and the conditional probability analyses (as section 2.6). Lines 175-185 were shifted from the previous section 2.2 to the new section 2.1

Thus, the new Sections are as follows:

2.1 Site Description

[revised manuscript text omitted]

* VOC sensitivities determined using VOC gas standards in calibration experiments
$ Observed mass accuracy for 1,3-Butadiyene and Propanenitrile were 21 mDa and 10 mDa, respectively
@ Corrected for the $^{13}C$ isotopologues of acetic acid

Lindinger, W., Hansel, A., and Jordan, A.: On-line monitoring of volatile organic compounds at pptv levels by means of proton transfer-reaction mass spectrometry (PTR-MS) medical applications, food control and environmental research, Int. J. Mass Spectrom., 173, 191–241, doi:10.1016/s0168-1176(97)00281-4, 1998.

A selection of the Pearson's coefficient R > 0.6 has been recommended by Norris et al. (2008) in the EPA PMF v3.0 user manual and more recently the same suggestion has been repeated in the user manual of the v5.0 (Norris et al. 2014). The recommendation has been generally adhered to by other authors using the same software e.g. Baudic et al., 2016.

We have now included the citation to Norris et al., 2008 and 2014 in the revised manuscript.

Norris, G., Vedantham, R.,Wade, K. S., Brown, S. G., Prouty, J. D., and Foley, C.: EPA positive matrix factorization (PMF) 3.0 fundamentals and user guide. Prepared for the US. Environmental Protection Agency, Washington, DC, by the National Exposure Research Laboratory, Research Triangle Park; Sonoma Technology, Inc., Petaluma, CA; and Lockheed Martin Systems Engineering Center, Arlington, VA, EP-D-05-004; STI-907045.05- 3347- UG, October, 2008.

Norris, G., Duvall, R., Brown, S., and Bai, S.: EPA Positive Matrix Factorization (PMF) 5.0: Fundamentals & User Guide, Prepared for the US, Environmental Protection Agency (EPA), Washington DC, by the National Exposure Research Laboratory, Research Triangle Park; Sonoma Technology, Inc., Petaluma, 2014.

Previously the sentence was (P5, L132-135):

"The model output of each bootstrap run is mapped onto the original solution using a cross correlation matrix of the factor contributions $g_{ik}$ of a given bootstrap run with the factor contributions $g_{ik}$ of the same time segment of the original solution using a threshold of the Pearson's correlation coefficient (R) > 0.6."

The revised sentence now reads as:

"The model output of each bootstrap run is mapped onto the original solution using a cross correlation matrix of the factor contributions $g_{ik}$ of a given bootstrap run with the factor contributions $g_{ik}$ of the same time segment of the original solution using a threshold of the Pearson's correlation coefficient (R) > 0.6, as suggested by Norris et al., 2008 and 2014."

Previously, P7, L219 was:

"To identify the uncertainty associated with the PMF solution, bootstrap runs were performed 100 times taking 96 hours as the segment length."

The revised sentence now reads as :

"To identify the uncertainty associated with the PMF solution, bootstrap runs were performed 100 times taking 96 hours as the segment length. This is slightly shorter than the recommended length based on the equation of Politis and White (2004), of 108 hours but represents a multiple of 24 hours and hence ensures each bootstrap run contains four full days' worth of data."

Politis, D. N. and White, H.: Automatic block-length selection for the dependent bootstrap, Econometrics Reviews, 23, 53–70, doi:10.1081/ETC-120028836, 2004.

**Specific comments: Section 2.1 This section needs some minor reorganization. A brief description of the PTR-ToF measurement should come first, then the description of the grab-sampling, then the PMF implementation. I suggest this because the PTR-ToF measurements, and the grab-samples, are referred to several times in the discussion of PMF; however, they had not yet been introduced.**

Done.

We have now reorganized the Materials and Methods section (section 2) as per the suggestion of the referee in the earlier comment, put new captions for the description of the PTR-ToF measurements and the grab-sampling and shifted the text to the relevant sections

**Line 127 I found this sentence very hard to parse. Perhaps you can break this down to provide a clearer explanation of the information provided by bootstrapping.**

Done.

Previously the sentence was (P5, L127):

"To ascertain the magnitude of random errors that can be caused due to the use of random seeds followed by the selection of the run with the lowest Q due to the existence of infinite solutions with different $g_{ik}$, $f_{kj}$ and $e_{ij}$ matrices but identical $Q = \Sigma_{i=1}^{n} \Sigma_{j=1}^{m} (e_{ij}/\sigma_{ij})^2$, bootstrap runs were performed."

The revised sentence now reads:

"Bootstrap runs were performed to ascertain the magnitude of random errors of the dataset (Norris et al. 2014, Paatero et al. 2014). Random errors can be caused due to the existence of infinite solutions with different $g_{ik}$, $f_{kj}$ and $e_{ij}$ matrices but identical $Q = \Sigma_{i=1}^{n} \Sigma_{j=1}^{m} (e_{ij}/\sigma_{ij})^2$."

Paatero, P., Eberly, S., Brown, S. G., and Norris, G. A.: Methods for estimating uncertainty in factor analytic solutions, Atmos. Meas. Tech., 7, 781–797, doi:10.5194/amt-7-781-2014, 2014.

**Lines 127-139 This section could benefit from literature citations describing the use of bootstrapping. I'd also like to see a citation supporting your assertion that a fraction <20% of unmapped factors indicates low random error.**

The statement that a solution in which >80% of the bootstrap runs are successfully mapped onto the same factor can still be considered a relatively stable solution can be found in Norris et al. (2014) in several places (e.g. discussion of table 9). The manual recommends that when the number of unmapped factors (= bootstrap factors that could not be mapped onto any one of the source profiles) is high, the users should carefully investigate which observations/outliers have a disproportionate influence on the factor profiles but could also explore lowering the threshold below 0.6. However, this is only of academic interest, since our model runs have no unmapped factors, while retaining the recommended threshold of 0.6. We have included a citation and elaborated further. Line 138ff now reads:

"The presence of a high fraction unmapped factors (> 20%) is a clear indication of large random errors (introduced by a few critical observations that drastically impact factor profiles) and should be investigated carefully (Norris et al. 2014). In our analysis, no unmapped factors were present."

**Lines 155-160 This explanation of rotational ambiguity is a little convoluted. Can you rephrase to make this easier to follow? See Ulbrich et al. (2009) as an example.**

Done.

Previously it was (P5, L155-160):

"In addition to the random error, the PMF model also has rotational ambiguity. There can be multiple solutions with a different factor profile for all factors for which the model will find a different local minimum of the residual matrix while determining the factor contribution matrix. This fact that different solution for $g_{ik}f_{kj}$ with the same sum of the scaled residuals $Q = \Sigma_{i=1}^{n} \Sigma_{j=1}^{m} (^{e_{ij}}/_{\sigma_{ij}})^2$ exist, is called the rotational ambiguity of the model."

In the revised manuscript, we have included the reference of Ulbrich et al., 2009 and modified the explanation of rotational ambiguity as follows:

"In addition to the random error, the PMF model also has rotational ambiguity (Ulbrich et al., 2009, Paatero et al. 2014). This rotational ambiguity is caused due to the existence of multiple solutions which have a Q similar to the solution produced by the PMF model but different factor profiles and factor contributions. Thus, the model will find different local minima of the residual matrix, while determining the factor contribution matrix ($g_{ik}f_{kj}$). The coexistence of different solutions for the factor contribution matrix ($g_{ik}f_{kj}$) with the same sum of the scaled residuals ($Q = \Sigma_{i=1}^{n} \Sigma_{j=1}^{m} (^{e_{ij}}/_{\sigma_{ij}})^2$) is called the rotational ambiguity of the

model."

Ulbrich, I. M., Canagaratna, M. R., Zhang, Q., Worsnop, D. R., and Jimenez, J. L.: Interpretation of organic components from Positive Matrix Factorization of aerosol mass spectrometric data, Atmos. Chem. Phys., 9, 2891-2918, 10.5194/acp-9-2891-2009, 2009.

**Line 192 What does it mean to be classified as a "weak" vs. a "strong" species?**

Weak species are considered to have a larger uncertainty. The specified uncertainty in the uncertainty file is tripled for weak species while for strong species the uncertainty is retained as is. This reduces the influence of the weak species on the factor profiles as it reduces the magnitude of the uncertainty scaled residual and hence the contribution to Q.

We have clarified this P6, L191-197 of the ACPD version of the manuscript (changes bold):

"Due to its erratic timeseries profile, HCN (m/z = 28.007) was classified as a weak species in the PMF input while all other ions were classified as strong species. **For weak species, the stated uncertainty is tripled, to reduce their impact on the scaled residual and hence Q.**"

**Line 193 The conversion from mixing ratio to mass concentration introduces additional variability due to meteorology. Additionally, higher molecular weight species now have more "pull" in the PMF solution, because their signals are higher. Do you see evidence for meteorological influence in the PMF solution? Any evidence of a bias towards explaining variability of heavy species, at the expense of light VOCs? Why did you choose to run PMF on the mass concentration data rather than mixing ratio data?**

We agree that the conversion of mixing ratio to mass concentration introduces additional uncertainty, however, the conversion was done using the relevant temperature, pressure and molecular weight and the uncertainty thus introduced is already accounted for by running the model with 5% measurement uncertainty. It is desirable and recommended to run the PMF after converting to mass concentrations (Norris et al. 2014). Only this conversion allows mass closure and hence the preparation of pie charts quantifying the source contributions (Figure 13; Section 3.3), which can be compared to emission inventories (Figure 14; Section 3.3).

It should be noted that for each observation the effect of each species residual on Q is scaled by the uncertainty of that very same species and observation. Therefore, higher mass does not at all result in a higher pull. A residual that is high compared to the measurement uncertainty of that specific observation exerts a pull. A residual that is negligible when compared to the measurement uncertainty of that specific observation has no influence on the output.

As described in Paatero et al. (2014), the highest "pull" on the PMF solution is due to species with a strong presence in some observations (in our case factor >4 enhancement) above the median mass loading, which is contrasted by low values (in our case less than 1/3 of the median) at other times. A strong presence-absence contrast most strongly defines source profiles.

It can be seen in the factor profiles and the sequence in which factors appear in the PMF output, that compounds with a low molecular mass are equally or more important in defining for the PMF solution space compared to higher mass species. Table S3 shows the percentage contribution of PMF derived factors obtained from constrained runs with 5-, 6-, 7-, 8- and 9-

Factors. The biogenic factor (which is primarily defined by biogenic isoprene and acetaldehyde emissions) splits from the mixed daytime factor in the 5-factor solution, and does so, before the traffic source (defined primarily by benzene, toluene, xylenes and trimethylbenzens) splits from mixed combustion sources in the six factor solution.

We have rephrased L193ff" "All the input data was converted from mixing ratios ppb to mass concentrations ($\mu g\ m^{-3}$) using the relevant temperature, pressure and molecular weight. The total measured NMVOC concentration was calculated by adding the mass concentrations of all measured NMVOCs and was classified as a weak species in the PMF input."B

The text now reads: "All the input data was converted from mixing ratios of ppb to mass concentrations ($\mu g m-3$) using the relevant temperature, pressure and molecular weight and the total measured NMVOC concentration was calculated by adding the mass concentrations of all measured NMVOCs. This conversion allows calculating the explained variability (Gaimoz et al. 2011) for the total VOC mass and comparing the results with emission inventories. The conversion does not introduce significant additional uncertainty and the variability induced by the temperature (average range observed was: 5-20 ℃), has largely been taken into account by running the model with a 5% extra modelling uncertainty. The total VOC mass was classified as a weak species in the PMF input (Norris et al. 2014)."

**Lines 200-220 On my first read-through, the presentation of the eight factors was very sudden and it wasn't at all clear to me how their identifications were derived, or that more information would be provided later in the text.**

This section mainly focuses on the description of the implementation of the PMF model to the VOC dataset and therefore in P7, L215-218 (of the ACPD version of the manuscript), only the names of the factors are mentioned which were derived from the diagnostics of the 8-factor solution of the PMF run. A detailed description of how the identification and attribution of these PMF derived 8-factors were performed is described in detail in section 3.1 of the Results and Discussion (section 3). However, we have now added a sentence after L218 (of the ACPD version):

"A detailed description for the identification and the attribution of the 8-factor solutions is provided later in section 3.1."

**It would be very helpful to see a plot of Q/Qexpected as a function of number of factors, and an additional plot showing what happens to each of your identified factors as the number of factors is changed (perhaps in the supplemental information).**

We have added a figure showing the % change in $Q/Q_{expected}$ when the number of factors is increased for all solutions starting from a 3 factor solution (in absolute terms $Q/Q_{expected}.<1$ even for a 3 factor solution). This dataset is highly unusual, due to the fact that wind speed and wind direction have a strong diel cycle throughout the campaign due to mountain meteorology. In this situation all traditional indicators for the quality of a PMF solution fail. We have also added a time series of the modelled mass and the measured mass, which clearly shows perfect mass closure even with a 3 factor solution already. A 3 factor solution

distributes compounds into the following factors: Factor 1: "higher during the first part of the campaign no diel profile", Factor 2: "higher during the second part of the campaign and higher at night and Factor 3 "higher during the second part of the campaign and higher during the day". Mass closure, however, does not mean this 3 factor solution is plausible and corresponds to real world source. It merely means that all mathematical quality indicators typically used fail for this dataset. Instead the plausibility of each possible solution must be carefully assessed keeping in mind the auxiliary information. However, the last unusually high drop in $Q/Q_{expected}$ is seen when the number of factors is increased to 8. Beyond that the relative change stays constant.

We have modified the statement in line 202 of the ACPD manuscript to make this clearer, instead of previously:
 "Based on the $Q/Q_{theoretical}$ ratio, the physical plausibility of the factors and the rotational ambiguity of the solution, an 8-factor solution was deemed to be the best for this dataset."
The text now reads"
"Based on the $Q/Q_{theoretical}$ ratio, the physical plausibility of the factors and constraints imposed by rotational ambiguity of the solution, an 8-factor solution was deemed to be the best for this dataset. For the data presented in this study, the $Q/Q_{theoretical}$ ratio is <1 even for a 3 factor solution with no physical plausibility and hence the absolute number does not help to decide the optimum number of factors. Supplementary Figure S2 shows clearly, that the number of factors has almost no impact on how well the total mass is reproduced by the model but the last distinct drop in the $Q/Q_{theoretical}$ ratio is seen when the number of factors is increased to 8."

[Figure]

**Figure S2.** Relative change in the Q/Qexpected ratio with change in factor number (top) and time series of the total measured VOC mass (grey filled) and the modelled VOC mass for different number of factors in the PMF solution (bottom).

Regarding the variation of PMF derived factors with number of factors, we have already provided a Table in the supplementary information (Table S2) we do not consider it necessary to replace this table with a Figure containing the same information. However, we have added a supplementary Figure S3 showing how factor profiles, the mass of each species explained by each factor and the factor contributions evolve from the 5 Factor solution to the 9 Factor solution.

In the factor profiles it can be seen that individual compounds move to the newly generated factor from several previous profiles when the number of factors is increased. More specifically, aromatics and several other compounds from a mixed combustion factor, brick kiln and residential burning factor move into two new profiles, the traffic factor and mixed industrial emission factor when the number of factors allowed is increased to 6. When the number of factors is increased to 7 Aldehydes and acids, previously distributed among the residential biofuel and waste disposal factor, brick kiln factor, mixed daytime and the biogenic factor move to the solvent evaporation factor. At the same time, the time series of the mixed industrial factor shows, that brick kiln emissions which got pushed into the mixed industrial factor while compromising the brick kiln source profile to accommodate more acetaldehyde and acetic acid, shift to the brick kiln factor. When the number of factors is increased to 8 several compounds previously accommodated in the mixed industrial, solvent evaporation and mixed daytime factor, most notably 1,3-butadiyne, shift to the new profile. The eight factor solution is the first one, where the factor contribution for residual night-time primary emissions in all daytime factors drop below 10 $\mu g/m^3$ for the full period.

[Figure]

**Figure S3a.** Evolution of the factor profiles of the eight sources identified and the 9[th] source which is considered to arise due to splitting of the brick kiln factor from the 5 Factor to the 9 Factor solution.

[Figure]

**Figure S3b.** Evolution of the percentage of the mass of each compound explained by the eight sources identified and the 9[th] source which is considered to arise due to splitting of the brick kiln factor from the 5 Factor to the 9 Factor solution.

[Figure]

**Figure S3c** Evolution of the factor contribution of the eight sources identified and the 9th source which is considered to arise due to splitting of the brick kiln factor from the 5 Factor to the 9 Factor solution.

Eight factors is a quite large number compared to PMF solutions reported in many other studies. Without strong supporting evidence for each factor, I find it hard to believe that PMF can robustly separate this many distinct sources. It is especially hard to believe when each factor, on average, accounts for 12.5% of signal – but you have stated that the overall measurement uncertainty is 20% (line 190). Any additional information that you can provide to show that an 8 factor solution is physically plausible would be very helpful in convincing the reader of your solution. This could

**include additional PMF diagnostics (as suggested above) or other information. For example, it seems that prior to running PMF, you had some idea of what important emission sources to expect, perhaps from your previous paper or from an emissions inventory (I see four listed in the introduction). Can you make a stronger connection between the a priori knowledge and the PMF solution?**

We thank the referee for this suggestion. It is true that we had some prior information and we have made a stronger connection in the text now. Before we performed the source apportionment using PMF model to the NMVOCs measured in the Kathmandu valley, we undertook a thorough analysis of the primary NMVOCs dataset as reported in our companion paper to this Special issue Sarkar et al., 2016. In addition information about local sources and emission activity periods (e.g. operation of brick kilns, change in meteorological conditions, was provided by co-authors. Finally we also relied on previous studies and information from existing emission inventories, for a good background. We summarize these below:

1) Biogenic emission sources (characterized by high daytime concentrations of isoprene emitted due to the presence of nearby forested areas). The biogenic factor is strongly defined by emissions from deciduous trees, which were present and responded quickly to changes in solar radiation during the first two weeks of the campaign (See Sarkar et al. 2016 and the response to reviewer #2 for a plot demonstrating how rapidly the vegetation responded to changes in cloud cover). Since the deciduous trees shed their leaves during the second part of the campaign and biogenic emissions from evergreen trees and needle leaved trees are much lower during that time, the overall penalty on Q for combining biogenic emissions with traffic emissions and residential burning into one "higher during the first part of the campaign" factor with no clear diel profile is low.

2) Biomass co-fired brick kilns emissions (characterized by the emissions of high concentrations of benzene from the nearby brick kilns and its excellent correlation with acetonitrile, presumably due to co-combustion of crop residue and other biomass). Brick kilns, were not operational during the first part of the campaign. This perturbation is sufficient to confidently separate their emissions from combustion emissions of other industrial units, which continued their operations throughout our study time period. At the same time all the industrial sources including brick kilns are spatially co-located and follow similar temporal patterns. Therefore, brick kilns and other industrial emissions can be combined into one "high during the second part of the campaign and high night-time emission factor" with a relatively low penalty on Q. The price to pay is that certain compounds that are high in the mixed industrial emissions but not in the brick kiln emissions are pushed into the residential burning factor.

3) Biomass/residential burning emission sources (characterised by the presence of the high concentrations of several oxygenated VOCs, acetonitrile and aromatic compounds) were expected to be an important source of VOCs based on a priori knowledge from the existing emission inventories (for the REAS inventory, this is the single most important source of VOCs).

4) Traffic emission sources (characterized by the presence of high concentrations toluene, C8- and C9-aromatics), were considered to be extremely important based on earlier studies conducted in the Kathmandu valley (Shrestha et al. 2013). However, since the measurement site was rarely downwind of Kathmandu valley during evening traffic rush hours, the traffic

factor is defined by a few strong plumes with mass loadings of ~120 μg/m³. As a consequence, the overall impact of the traffic source on Q is low and the emissions from this source can be combined with the residential burning factor with a relatively low penalty on Q.

5) Mixed daytime/photochemical sources are characterized by the presence of compounds such as isocyanic acid during daytime which is produced as a result of photooxidation of precursor amides. This source was already demonstrated to be an important source for several compounds in the companion paper. Since photochemical formation of secondary pollutants is clearly important in the Kathmandu valley it is highly desirable to segregated this secondary pollution from primary emissions which proves to be challenging in a less than 8 factor solution.

6) Contribution from industrial sources located in nearby industrial estates were expected and could be separated from brick kilns thanks to the fact that brick kilns were closed during the beginning of the campaign.

7) Solvent usage is an important source of VOCs according to several emission inventories and the single most important source according to the EDGAR emission inventory. Due to the foggy/hazy conditions at night this source has a very peculiar time series. Soluble compounds tend to partition into the fog/haze aqueous phase at night and rapidly partitions into the gas phase during morning hours. This is particularly true for compounds with a high temperature dependence of their solubility in water. This, however, means non-water soluble solvents and water soluble solvents do get separated by the PMF and the later land in the mixed industrial and unresolved industrial factor.

While based on the preliminary information we had prior to initializing the model it seemed best to run the model with only 7 factors the raw time series data (input data) strongly supports an 8 factor solution. We have added the following text to clarify this:

[revised manuscript text omitted]

**Line 204 You discarded solutions with 7 or fewer factors because there appeared to be "mixing" of sources. But, this could also be due to rotational ambiguity. Did you attempt to unmix solutions with ≤7 factors by exploring FPEAK, for example?**

FPEAK does not help removing primary emissions from the daytime profiles with 7 factors only. The constraint mode achieves superior results when the two options are compared for an equal number of factors. It should be noted that FPeak (available in the previous version of the PMF) and the constraint mode (new feature of the 5.0 version) are two alternate rotational tools which cannot be used in combination. We opted for the constraint mode, which allows us to use source profile fingerprints from samples collected at the source to refine the solution. The constraint mode performs is superior , as it exploiting rotational ambiguity to push the solution into a physically more realist space using pre-existing knowledge and the user decides how much price in the form of a "higher than the local minimum Q" he/she is willing to pay for a solution that corresponds better to the real world. FPeak simply explores how much the solution can be changed due to rotational ambiguity and the user usually goes for the solution with minimum Q. It has recently been recognized that a minimum Q represents the mathematical minimum but the same does not correspond to the most plausible real world solution (Paatero et al. 2014). We have added a statement that FPeak and the constraint mode are two alternate rotational tools and cannot be combined with each other to line 222 of the ACPD version, which reads:

"The constraint mode is a new rotational tool introduced in the 5.0 version of the EPA PMF as an alternative to the FPeak module. The constraint mode allows to exploit the rotational ambiguity of the model to push the PMF solution into a physically more realistic space. It uses pre-existing knowledge such as source fingerprints, source emission ratios or activity data. We found that when the two modules were compared for an equal number of factors the constraint mode performance was superior to the FPeak module."

**Line 226 Can you provide a scatterplot showing $R^2$ between each factor time series, vs $R^2$ between each factor profile?**

We have added the plot to this response, however, we are showing R vs R, since $R^2$ can have a positive 1 both for perfect correlation and anti-correlation. The information provided by this plot seems to be more difficult to interpret compared to the information provided by the plot presently available in the supplementary material so we have retained the old plot.

[Figure]

 **It isn't clear to me if you are discussing correlation between time series or profile here, and it would help to see the "whole picture".**

We are discussing the correlation between the time series. We have now clarified it…

"The original model output showed positive correlations between **the factor contribution time series**  **of** the biomass co-fired brick kilns and mixed industrial emissions ($R^2 = 0.27$) f**actor** as well as the residential biofuel use and waste disposal factor with traffic factor ($R^2 = 0.42$)."

**Lines 237-240 Not sure I understand this. Is "mixed daytime" a photochemistry tracer? Why wouldn't solvent evaporation contain acetonitrile or aromatics? Both acetonitrile and aromatics are commonly found in solvents. Why use a ratio to acetic acid as the nudging control?**

Yes, "mixed daytime" is primarily a photochemistry tracer, although we were not able to completely remove primary emissions from the photochemistry source profile and, therefore, continue to call it "mixed daytime".

The primary problem is that once resolved as separate, the solvent evaporation factor with its sharp solubility driven peak is not at all amenable to accepting aromatic compounds, which are not water soluble, into its factor profile. The factor profile contains no acetonitrile and only <1% contribution from other aromatic compounds to start with. Instead, the PMF prefers to deposit these compounds with the photochemistry source or biogenic emissions. If constraints are only placed on these two factors, the PMF will simply execute a complete factor swap while running the bootstrap runs. It will shift all compounds that were in the biogenic factor to the solvent evaporation factor (which has no problems in accepting the constraints on aromatic compounds with no penalty on Q) and all the compounds that were in the solvent evaporation factor to the biogenic factor without making changes to the source profile of the biogenic emissions during the constraint runs. Primary emissions can only be restricted in the biogenic and mixed daytime factors by placing constraints on all daytime factors and constraining with the ratio to acetic acid as well, rather than with ratios to isoprene only. The reason for the second constraint is, that when a compound is completely removed from a factor profile, the constraining emission ratio no longer applies (as the constraining equation is not defined for 0 in the denominator). If constraints are placed on isoprene alone, but for all daytime factors, the model ends up remove isoprene completely from the mixed daytime factor during the constraint run. This allows the model to shift all the

aromatics all other primary emissions into the photochemistry factor, rather than shifting them to one of the combustion sources or the solvent evaporation factor which costs a higher Q.

**Figure 2 To which axis do the gray bars and red lines-and-markers belong, respectively?**

In Figure 2, the left axis corresponds to the grey bar and the right axis corresponds to the red lines and markers, respectively.:

"Figure 3 represents the factor profiles of all the eight factors resolved by the PMF model in which grey bars (left axis) indicate the mass concentrations and red lines with markers (right axis) show the percentage of a species in the respective factor."

**Line 370 You are interpreting m/z 69 $C_5H_8H^+$ as isoprene, correct? I suggest that this ion mass is actually a cycloalkane or alkane fragment, which seems far more plausible for vehicle exhaust. See Gueneron et al. (2015).**

We thank the reviewer for pointing us to Gueneron et al., 2015 this fragment is indeed a potential explanation for $C_5H_8H^+$ in vehicle exhaust, although Borbon et al. 2001 (already cited in the paper) identified isoprene in the emissions of petrol vehicles that were not equipped with catalytic converter using GC FID not PTR-MS and found the emission factors were equivalent to those of pentenes an butenes. Considering the old fleet plying in Kathmandu valley, isoprene is a plausible contributor to the traffic source. Even so, if we consider that the fragmentation of cycloalkanes and cycloalkenes should also result in product ions at m/z 111 and/or m/z 125 and the signal at those masses at ~135 Td should be e above 0.2 ppb. However, in the observed mass spectra, there was no significant signal at these m/z values. Therefore, we conclude that isoprene is the more plausible assignment. As this is an important point, we have included this discussion in the revised version along with a citation to Gueneron et al., 2015.

We inserted the following text (insertion bold) into line 370f: "Few previous studies **employing GC-FID** have reported traffic related sources of isoprene in urban areas (Borbon et al., 2001; Hellèn et al., 2012)" and have added to line 372ff "**A recent study suggested m/z 69 $C_5H_8H^+$ could also result from the fragmentation of cycloalkanes and cycloalkenes (Gueneron et al., 2015). Fragmentation of these compounds should also result in product ions at m/z 111 and/or m/z 125 and the signal at those masses at ~135 Td should be above 0.2 ppb. However, in the observed mass spectra, there was no significant signal at these m/z values. Therefore, we conclude that isoprene is the more plausible assignment.**"

**Line 477 Why would solvent evaporation correlate with the rate of change of temperature/sunlight, and not directly with temperature?**

Assuming that the gas phase is in constant equilibrium with the aqueous phase at all times, mixing ratios should correlate with the change in temperature and not the absolute

temperature. Considering a case where the water solubility of a compound or the saturation vapour pressure changes by a fixed factor for a fixed temperature difference. The spike in the gas phase mixing ratios of the compound would be sharper, if the temperature change occurred in a shorter period of time and the increase would be more gradual, if the same temperature changes occurred more slowly. Also the increase of the mixing ratios during daytime is counteracted by the dilution effect. When the rate of the temperature increase per unit time decreases in the late morning, the compounds no longer partition into the gas phase fast enough to overcome the dilution effect, hence the mixing ratios start dropping when the rate of change slows, even before it becomes negative.

We have shifted the following text from L536 to L477 to make the reasons more clear and also added the citations for the change in the solubility.

"However, the change of the saturation vapor pressure for a temperature change from 5$^o$C to 20$^o$C for the dominant compounds (acetaldehyde and acetic acid) present in the solvent evaporation factor is small (less than a factor of 1.3; Betterton and Hoffmann (1988); Johnson et al. (1996)) and, therefore, does not account for the observed magnitude of increase (by a factor of ~5) from 06:00 - 09:00 LT. Instead, the temperature dependence of the solubility of these compounds in an aqueous solution would explain a change of this magnitude." This information is already provided in P21, L536 – P22, L546 of the ACPD version of the manuscript we have now shifted it to Line 477. The temperature change drives the compound from the aerosol aqueous phase into the gas phase.

**Lines 545-548 Can you state (or reiterate, possibly I missed this above) why it cannot be that the solvent evaporation and unresolved industrial factors are "split" from a single source by the PMF? This section also seems fairly complex and highly speculative. Can you cite an example where such a situation has been shown to occur?**

We are aware of previous papers exploring the impact of atmospheric conditions on the PMF output (e.g. Yuan et al. 2012), however, we are not aware of any other case where the gas phase mixing ratios were affected by the presence of a large aerosol aqueous phase. In previously reported studies the complications were caused by photochemistry. The two factors cannot be combined because the two correlate only during the day (R=0.55) and not during the night (R=0.29). When day and night are clubbed together R drops to 0.42. At night the solvent/evaporation factor anti-correlates with RH (R= -0.59) while the unresolved industrial factor has only a mild positive correlation with RH (R=0.29). During the day the solvent/evaporation shows the highest correlation with ΔT (R=0.64) while the unresolved industrial factor shows no significant correlation with ΔT (R=0.28). The raw data, now added as Table S5, also suggests against combining these two factors. The time series of measured acetaldehyde and acetic acid show a rather weak correlation with 1,3-butadiyne and methanol (R<0.54). On the other hand, the measured time series of 1,3-butadiyne and methanol correlates extremely strongly (R=0.9), indicating there is a strong and unique common source which causes sharp spikes in these two compounds which has very different emission ratios of 1,3-butadiyne to acetaldehyde, acetic acid and formic acid compared to the solvent evaporation factor (which is not a significant source of 1,3-butadiyne and methanol). The fact that the correlation of 1,3-butadiyne with ethanol, the defining compound of the mixed

industrial factor, ethanol, is equally poor, speaks against combining the mixed industrial factor with the unresolved industrial factor.

The referee is correct that this section was poorly supported by data and we now address this valid concern by adding supplementary table S5 and the following text which replaces the original:

 "While the correlation of the solvent evaporation factor with the unresolved industrial factor during daytime seems to suggest the two should be combined into one factor profile, several facts suggest against it. Firstly, the two do not correlate at night since the unresolved industrial factor shows a mild positive correlation rather than anti-correlation with RH at night (R=0.29) and no strong correlation with $\Delta$T during the day (R=0.28). Secondly, the raw time series of 1,3-butadiyne and methanol (Supplementary table S5) correlates extremely strongly (R=0.9), indicating there is a strong and unique common source which causes sharp spikes in these two compounds. The fact that the time series of 1,3-butadiyne correlates poorly with acetaldehyde, acetic acid and formic acid indicates that the solvent evaporation factor (which is not a significant source of 1,3-butadiyne and methanol), has very different emission ratios of 1,3-butadiyne to acetaldehyde, acetic acid and formic acid compared to the unresolved industrial emissions factor to explain the raw data. The fact that the time series of 1,3-butadiyne correlates equally poorly with that of ethanol, the defining compound of the mixed industrial factor, suggests against combining the mixed industrial factor with the unresolved industrial factor. It, therefore, seems, that the unresolved industrial factor is related to primary emissions from a distinct source, while the source profile and diel cycle of the solvent evaporation factor may be strongly confounded by meteorology and chemistry. Confounding factors have been reported to affect PMF solutions previously (Yuan et al. 2012)".

**Lines 600-610 This also seems to point to a "splitting" of a single source into the unresolved industrial and solvent evaporation factors.**

In the context of our study a "single source" would be a specific industrial point source or a specific well constrained sector which can be targeted by policy makers to reduce pollution. The fact that the industrial units responsible for the emissions associated with the solvent evaporation factor and the unresolved industrial factor are likely located in the same two industrial estates does not necessarily mean, the same plant or even the same sector is to be blamed for both types of emissions. The source profile of the unresolved industrial emissions is very specific and points towards plastic/adhesives/pharmaceutical industries. The source profile of the solvent evaporation factor is so strongly confounded by meteorology, that the origin of the emissions cannot be determined with great confidence. However, the fact that the acetic acid and acetaldehyde mass attributed to this factor is primarily distributed between three different combustion sources (brick kiln, residential fuel use and waste disposal and mixed industrial) when the number of factors is reduced to 6 (Figure S3a & b) indicates that multiple (combustion) sources contribute to the primary emission. The mass of the unresolved industrial emissions, on the other hand, gets distributed between the mixed industrial and mixed daytime factor when the number of factors is reduced to 7 (Figure S3a&b). The removal of a significant fraction of the mass of certain compounds from the mixed industrial into the unresolved industrial factor is accompanied by an almost complete

separation of the conditional probability functions of these two factors. This means genuine sources are split from each other when the number of factors is increased to 8 and except for a conditional probability function pointing to two specific industrial estates and a late morning peak in emissions, the solvent evaporation and the unresolved industrial factor do not have much in common.

**Figure 17 Can you also include the time series of the total VOC mass loading?**

Done.

We have now included the timeseries of the total VOC mass loading in the revised Figure 17 (now 18) of the manuscript

[Figure]

**Line 818 Many of the oxygenated VOCs are direct emissions from solvents, industry etc (Figure 16). So I do not think it is correct to say that photochemically produced VOCs are a dominant source of O₃ potential, especially when Figure 18a shows that the mixed daytime source contributed only about 5%.**

We thank the reviewer for drawing attention to the confusion caused by the choice of words at Line 818. We completely agree that OVOCs have considerable anthropogenic sources in the Kathmandu Valley too. In fact this point was made strongly at Line 821 just three lines after L818 of the original submission, and in L818 we were only trying to make the point that without the PMF analyses, measurements of OVOCs and isoprene, which in several ambient environments are primarily controlled by photochemistry and biogenic sources, could have led to the premature assumption that these natural sources are more important for the daytime ozone formation potential in the Kathmandu Valley, whereas in fact anthropogenic sources

are more important by collectively contributing 70% to the mass loading as noted in Line 821 of original submission .

In the revised version, we rephrased L 818 as follows to avoid potential confusion by adding the word "presumptuously" as follows::

"The distribution of the daytime $O_3$ production potential obtained from the measurements (Figure 19b) shows that 78% of the total daytime O3 production potential was due to the contribution from isoprene and oxygenated NMVOCs which could *presumptuously* indicate dominance of biogenic emissions and photochemistry in the Kathmandu Valley even in the winter."

**Lines 847-848 Can you clarify how this is related to a result of your work.**

Done,  The paragraph now reads:

"Speciation of NMVOCs in the emission inventory for Nepal only includes compound classes (e.g. alkanes, alkenes etc.) and not specific compounds. This imposes certain limitations while comparing emission inventories with the compounds measured in our study. However, the existing emission inventories …"

**Conclusion The conclusion is heavily weighted towards comparison with the emission inventories. While this is an important result, it is not the only finding discussed in the paper. The conclusion could be improved by an assessment of the major findings related to the source contributions to different categories of VOCs, specific VOCs, and $O_3$ and SOA formation potential.**

Done.

We have now included a paragraph listing all the other important findings of this study in the conclusion.

"Eight different NMVOC sources were identified by the PMF model using the new "constrained model operation" mode. Unresolved industrial emissions (17.8%), traffic (16.8%) and mixed industrial emissions (14.0%) contributed most to the total measured NMVOC mass loading while biogenic emissions (24.2%), solvent evaporation (20.2%), traffic (15.0%) and unresolved industrial emissions (14.3%) were the most important contributors to the ozone formation potential. Biomass co-fired brick kilns and traffic contributed approximately equally to the secondary organic aerosol (SOA) production (28.9% and 28.2%, respectively), while the most important contributors to the mass loadings of carcinogenic benzene were brick kilns (37.3%), unresolved industrial (17.8% and mixed industrial (17.2%) sources. Photo-oxidation (mixed daytime factor) contributed majorly to two newly identified ambient compounds namely, formamide (41.1%) and acetamide (36.5%) along with their photooxidation product isocyanic acid (40.2%).

**Minor comments (typographical corrections):**

**Lines 140-145 Put all verbs in present tense for consistency ("will provide" → "provides")**

Done

**Line 203 "Fewer" than 7 factors**

Done

**Line 235 "constraints"**

Done

**Line 289 "FCBTBK": what does this acronym stand for?**

FCBTBK stands for fixed chimney bull's trench brick kiln. This has already been mentioned in P8, L245 of the ACPD version of the manuscript.

**Figures 15, 16 Can you include the explanation for the different source acronyms (e.g. "MD, SE, UI") in the caption.**

Done.

We have now added the full form of all the acronyms for Figure 15 to Figure 17 in the revised version of the manuscript.

---

## Author Comment (AC2) · 23 May 2017

**Source apportionment of NMVOCs in the Kathmandu Valley during the SusKat-ABC international field campaign using positive matrix factorization**

by Chinmoy Sarkar et al., 2017 (ACPD)

**REFEREE 2:**

**General comment: The manuscript shows results of a source apportionment study of NMVOCs measured by PTR-TOF-MS in the Kathmandu Valley in Nepal during winter 2013. Positive matrix factorization analysis was conducted to identify possible emission sources for 37 m/z measured by PTR-MS. The sources were identified from the chemical fingerprint of each PMF factor and their diurnal profiles. Conditional probability functions plots were used to determine the directions of the sources and attribute the chemical emissions to specific spatial areas in the region and specific activities. The sources found by the authors through PMF were compared with the results of current emission inventories used for Nepal, which, in contrast to the authors results, rely on sources emission factors measured in other regions of the world and are not supported by in-situ collected measurements. Sources and species contributions differ among the authors results and the current inventories as well as between different inventories. Finally, the atmospheric impact as daytime ozone production and SOA formation based on the measured compounds and PMF sources contributions is briefly discussed. I found the manuscript very interesting, of high quality and of high impact as it presents several new findings which can help mitigating the emissions in the region under study. The presented topic also follows in the scope of ACP. The article is overall well written, and results are presented clearly with figures and tables. I highly recommend the manuscript publication, once these specific comments have been addressed:**

We thank the referee for the kind words appreciating the importance of the work and for highly recommending the manuscript for publication in ACP. We have found several of Referee 2's specific comments very helpful and these are now reflected in the revised submission (changes are specified in replies and manuscript version with "tracked changes" given at the end of the responses here).

**Specific comments:**

**-It is a bit confusing how the methods section is presented. There should be a small section introducing the measurement site, the PTR-MS data used, and the grab samples, before any PMF discussion. This would be helpful to follow better section 2.2 and support the nudging tool. Could you list the m/z from PTR-MS you used for implementing the PMF and why? Could you also provide some references of the nudging procedure?**

Done.

We have now reorganized the Materials and Methods section (section 2) as per the suggestion of both the referees.

We have now included a column to Table S1 of the supplementary information to show the m/z ratios corresponding to the NMVOCs used for PMF. The detailed description concerning selection of these NMVOCs for the PMF run has now been added to the revised Materials and Methods section (Section 2).

The nudging procedure described in this work was performed using the priori knowledge of the emission sources in the Kathmandu valley and the emission ratios (ERs) obtained from the analysis of the grab samples collected from the point sources. This is the first ever study to use such nudging procedure to obtain robust solution using PMF. An earlier work of Baudic et al., 2016 has mentioned the need of using the nudging procedure/constrains using priori knowledge of the emission sources and ERs obtained from point sources to obtain robust solution using PMF but did not implement the same.

**-PTR-TOF-MS usually provides an unambiguous identification of chemical species, however, it would be interesting to know briefly, the operational settings of PTR-MS, which m/z were selected for running the PMF and how the m/z were attributed to the chemical compounds. Were the grab samples measured with the same PTR-MS? Could you provide some information about these data: m/z selected and how the compounds were identified. Line 331, could you provide the standard deviation for the signal stability?**

We have now included a section on PTR-TOF-MS measurements in the revised manuscript (Section 2.2 of the revised manuscript) that briefly discusses about the operational settings of the PTR-TOF-MS, sampling of ambient air and the identification procedure of the NMVOCs. The m/z ratios to the corresponding NMVOCs used for the PMF run is now provided in Table S1 of the supplementary information.

No, the grab samples were measured with a PTR-QMS which is installed at IISER Mohali, India. The analytical details, calibration procedure and information regarding the identification of NMVOCs using this PTR-MS are available in Sinha et al., 2014.

For the grab samples we only reported 7 compounds which we have tested to be stable in glass flasks. These compounds are: acetonitrile, benzene, toluene, sum of C8 aromatics, sum of C9 aromatics, styrene and naphthalene.

The zero air background for the m/z reported was $0.04\pm0.05$ ppb, $0.04\pm0.04$ ppb, $0.04\pm0.06$ ppb, $0.07\pm0.08$ ppb, $0.10\pm0.11$ ppb, $0.02\pm0.06$ ppb and $0.02\pm0.05$ ppb for acetonitrile, benzene, toluene, sum of C8 aromatics, sum of C9 aromatics, styrene and naphthalene, respectively. The concentration range in the grab samples was $4\pm0.3$ to $323\pm8$ ppb for acetonitrile, $27\pm4$ to $339\pm19$ ppb for benzene, $32\pm5$ to $150\pm14$ ppb for toluene, $40\pm6$ to $113\pm8$ ppb for C8 aromatics, $33\pm6$ to $62\pm12$ ppb for C9 aromatics, $11\pm1.3$ to $95\pm17$ ppb for styrene and $11\pm1.5$ to $64\pm9$ ppb for naphthalene.

We have now included this information in the Section describing the grab sampling and included a citation to the article that details the storage stability and validation of the glass flask sampling procedure and thank the referee for the excellent suggestion.

Citation:

Chandra, P., Sinha, V., Hakkim, H. Sinha, B.: Storage stability studies and field application of low cost glass flasks for analyses of thirteen ambient VOCs using proton transfer reaction mass spectrometry, International Journal of Mass Spectrometry, https://doi.org/10.1016/j.ijms.2017.05.008, 2017.

**-It would be interesting to provide some details about the calculations for ozone and SOA formation, you could do this with a short section in the methods after section 2.4.**

The ozone formation potential of individual NMVOCs was calculated as described by the following equation (Sinha et al., 2012):

$$Ozone\ production\ potential = (\Sigma k_{(VOC_i+OH)}\ [VOC_i]) \times OH \times n$$

For the ozone production potential calculation, the average hydroxyl radical concentration was assumed to be [OH] = $1 \times 10^6$ molecules cm$^{-3}$ with n = 2 and only data pertaining to the mid-daytime period were considered (11:00 - 14:00 LT).

This information is now included in Section 2.7

"The ozone formation potential of individual NMVOCs was calculated as described by the following equation (Sinha et al., 2012):

$$Ozone\ production\ potential = (\Sigma k_{(VOC_i+OH)}\ [VOC_i]) \times OH \times n$$

For the ozone production potential calculation, the average hydroxyl radical concentration was assumed to be [OH] = $1 \times 10^6$ molecules cm$^{-3}$ with n = 2 and only data pertaining to the mid-daytime period were considered (11:00 - 14:00 LT)."

Secondary organic aerosol (SOA) production was calculated using the concentrations and the known SOA yields for benzene, toluene, styrene, xylene, trimethylbenzenes, naphthalene and isoprene (Ng et al., 2007; Chan et al., 2009; Yuan et al., 2013; Kroll et al., 2006). SOA yield of a particular NMVOC depends on the NO$_x$ conditions and Pudasainee et al. (2006) previously reported NO$_x$-rich conditions in the Kathmandu valley. Therefore, SOA production was calculated by using reported SOA yield at high NO$_x$ conditions according to the following equation:

$$SOA\ production = [VOC_i] \times SOA\ yield\ of\ VOC_i$$

This information is now included in Section 2.7

"SOA yield of a particular NMVOC depends on the NO$_x$ conditions and Pudasainee et al. (2006) previously reported NO$_x$-rich conditions in the Kathmandu valley. Therefore, SOA production was calculated by using reported SOA yield at high NO$_x$ conditions according to the following equation:

$$SOA\ production = [VOC_i] \times SOA\ yield\ of\ VOC_i"$$

Pudasainee, D., Sapkota, B., Shrestha, M. L., Kaga, A., Kondo, A., and Inoue, Y.: Ground level ozone concentrations and its association with NO$_x$ and meteorological parameters in

Kathmandu Valley, Nepal, Atmos. Environ., 40, 8081–8087, doi:10.1016/j.atmosenv.2006.07.011, 2006

**-Figure 2. The contribution of propyne compared to isoprene for the biogenic factor is quite high, can you comment it?**

Since the source fingerprint of the primary source (traffic) is determined by night time emissions and the traffic factor profile during the daytime is different due to photochemical loss (between the source and the receptor downwind). As a consequence, some of the co-emitted compounds (in particular those on which we placed no specific constraint) remain in the biogenic source profile even after constraints are imposed to remove combustion derived isoprene and other associated primary emissions such as propyne.

**-Figure 3. How do you explain the higher background and general higher peaks during the first part of the campaign? L. 370: Could you provide more information about isoprene emission from traffic? Could you have any interference on the PTR-MS m/z attributed to isoprene?**

The higher peaks during the first part of the campaign are due to emissions from deciduous trees which shed their leaves during the latter part of the campaign. The highest peaks during this period occur during the daytime and not at night. We have provided additional references for isoprene emissions from traffic and have discussed them in more detail also in response to reviewer 1.

During the measurement period, significant isoprene concentrations (~ 0.5-2 ppb) were observed during evening and night time which are likely from biomass combustion and traffic emission sources (Sarkar et al., 2016) as the evening and night time isoprene has a strong correlation with vehicular emission tracer toluene. The following figure (Figure S9 of the supplementary material) shows an illustrative day's (18 January 2013) isoprene data against solar radiation. It can be observed from the figure that the daytime isoprene emission correlates very nicely with solar radiation which indicates biogenic emission while during evening hours and night time, isoprene showed high peaks that show good correlation (r>0.9) with toluene.

[Figure]

The issue of possible interferences to the isoprene signal has already been discussed above while addressing the comment of referee 1 and revisions have been made as outlined in the reply to reviewer 1's comment.

**Could be there a connection between oxidation products of traffic emission and the unresolved industrial emissions, as for mixed daytime emissions as oxidation products from biogenic emissions?**

The traffic factor is dominated by the contribution of toluene and higher aromatics (C8- and C9-aromatics). The oxidation of all these aromatics produces phenols and cresols. However, we did not observe phenols and cresols above 200 ppt in the Kathmandu valley. The unresolved industrial emissions factor is dominated by propene, propyne, methanol, acetone, acetic acid, formic acid and 1,3 butadiyne. Propene, propyne and 1,3 butadiyne cannot be formed due to oxidation of higher aromatics. Consequently the methanol, acetone, acetic acid and formic acid found in the same factor profile cannot be from photo-oxidation either. Furthermore, a bimodal diel profile as observed for the unresolved industrial emissions profile is not characteristic of photochemically emitted compounds.

The mixed daytime emissions profile is dominated by nitrogen containing compounds, most notably isocyanic acid, and its precursors formamide and acetamide. In addition, the profile contains photochemically formed methanol, acetone, acetaldehyde, formaldehyde, formic acid and acetic acid. One of its features is, that the mass loadings of the photo-oxidation products present in the mixed daytime increases after the brick kilns resume operation. The biogenic emissions, on the other hand, decrease in the second half of the campaign when the deciduous trees shed their leaves. As a consequence, the primary oxidation products of isoprene, MVK+MACR (methyl vinyl ketone + methacrolein) and MEK (methyl ethyl

ketone). MVK+MACR are associated with the biogenic emission factor itself and do not enter the photo-oxidation factor.

**-Why biogenic emissions are higher during the first part of the campaign? Were temperature and solar radiation also higher for this part of the campaign?**

Primarily because deciduous trees shed their leaves in early January (Sarkar et al., 2016) and no longer contribute to biogenic emissions during the second part of the campaign. We do not have measurements of temperature and solar radiation from the first part of the campaign due to a software glitch but in general conditions were warmer in the first part with reduced fog relative to the second part of the campaign.

**-Section 3.2 would be easier to follow with a map of the measurement site and mentioned cities, industrial estates and forests.**

Done.

We have now added a map of the measurement site to the revised manuscript (Figure 1).

**-Section 3.3, The differences between the current inventories used in Nepal are briefly mentioned in the text, however, it would be interesting to write a few lines at the beginning of this section to introduce the inventories and on which data and assumptions they are based on. Is EDGAR v4.2 also considered for the winter season?**

For EDGAR v4.2 spatially resolved seasonal data is not available. We have mentioned in the text and in the figure that the EDGAR v4.2 emission inventory are for the year 2008, while REAS 2.1 emissions are for December and January 2008. To make this clearer we have inserted "full" before "year" and added the following text in the paragraph on the EDGAR inventory: "EDGAR v4.2 inventory provides only spatially resolved data, not seasonally resolved data."

**- Lines 786-808, Did you compare your NMVOCs data with the wind directions? How was the wind direction affecting the sources emissions captured at the measurements site?**

The conditional probability function (CPF) plots shown in Figure 12 shows the wind directional dependency of different source categories reported in this study. The figure is discussed in Section 3.2.

Lines 786-808 describe the time series of the total VOC mass. The raw NMVOC data and its dependency on wind direction was analysed in Sarkar et al. (2016) already. As stressed in Sarkar et al. (2016) and in the materials and methods section of the current paper, during this time of the year, wind direction and speed in the Kathmandu valley usually followed predictable diurnal cycles. Such behaviour is typical for a site heavily influence by mountain meteorology. Hence the changes in the source strength of emission sources are not caused by

changes in the wind direction, between the first and the second part of the campaign. They are due to genuine changes in the activity/emission strength.

**-Line 815 and 840, please give the equations used for O₃ and SOA formation with respective references. It is not easy to understand figure 18 without any specification on the compounds used for each pie chart. Were the measured data used for pie b) the same data sets used to run the PMF?**

Done. We have now included the equations used to calculate the $O_3$ and SOA formation in the materials and methods section of the revised manuscript.

Yes. The measured data used in Figure 19.b) is the same data set used to run the PMF model. As suggested by reviewer 1 we have improved the discussion of this figure.

**-Line 828-832, much information is provided, please rephrase the period. Conclusions: Please include a short summary of the main findings here.**

Done.

Earlier the sentence was:

"Based on measured methane and 63 non methane hydrocarbon measurements in the city of Lahore which is much larger and by all indications more polluted than Kathmandu (Barletta et al. 2016)) the authors reported a maximum contribution of about 14% due to all alkanes including methane to the total measured OH reactivity."

We have now modified this as follows:

"For the city of Lahore, Barletta et al.2016 , reported the maximum contribution of methane and 63 non methane hydrocarbons to the total measured OH reactivity as 14%. Lahore, is much larger and by all indications more polluted city than Kathmandu."

Done. We have now included a short summary of the main findings in a paragraph in the conclusions as mentioned while addressing the comments of referee 1.

**Technical comments: -Some acronyms are not explained, or only explained once in the whole manuscript. Could you also provide the extended form of all acronyms used for tables and figures in their captions?**

Done.

The extended form of the acronyms are now provided in the figure and table captions/footnotes.

**-L. 698, ca. 30%.**

Done